# Efficient Adversarial Attacks on Online Multi-agent Reinforcement Learning

**Guanlin Liu**     **Lifeng Lai**
Department of Electrical and Computer Engineering
University of California, Davis
One Shields Avenue, Davis, CA 95616
`{glnliu, lflai}@ucdavis.edu`

## Abstract

Due to the broad range of applications of multi-agent reinforcement learning (MARL), understanding the effects of adversarial attacks against MARL model is essential for the safe applications of this model. Motivated by this, we investigate the impact of adversarial attacks on MARL. In the considered setup, there is an exogenous attacker who is able to modify the rewards before the agents receive them or manipulate the actions before the environment receives them. The attacker aims to guide each agent into a target policy or maximize the cumulative rewards under some specific reward function chosen by the attacker, while minimizing the amount of manipulation on feedback and action. We first show the limitations of the action poisoning only attacks and the reward poisoning only attacks. We then introduce a mixed attack strategy with both the action poisoning and the reward poisoning. We show that the mixed attack strategy can efficiently attack MARL agents even if the attacker has no prior information about the underlying environment and the agents' algorithms.

## 1   Introduction

Recently reinforcement learning (RL), including single agent RL and multi-agent RL (MARL), has received significant research interests, partly due to its many applications in a variety of scenarios such as the autonomous driving, traffic signal control, cooperative robotics, economic policy-making, and video games [Silver et al., 2016, Brown and Sandholm, 2019, Vinyals et al., 2019, Berner et al., 2019, Shalev-Shwartz et al., 2016, OroojlooyJadid and Hajinezhad, 2019, Baker et al., 2020, Zhang et al., 2021a]. In MARL, at each state, each agent takes its own action, and these actions jointly determine the next state of the environment and the reward of each agent. The rewards may vary for different agents. In this paper, we focus on the model of Markov Games (MG) [Shapley, 1953]. In this class of problems, researchers typically consider learning objectives such as Nash equilibrium (NE), correlated equilibrium (CE) and coarse correlated equilibrium (CCE) etc. A recent line of works provide non-asymptotic guarantees for learning NE, CCE or CE under different assumptions [Sidford et al., 2020, Zhang et al., 2020a, Bai and Jin, 2020, Xie et al., 2020, Liu et al., 2021, Jin et al., 2021, Mao and Başar, 2022].

As RL models, including single agent RL and MARL, are being increasingly used in safety critical and security related applications, it is critical to developing trustworthy RL systems. As a first step towards this important goal, it is essential to understand the effects of adversarial attacks on RL systems. Motivated by this, there have been many recent works that investigate adversarial attacks on single agent RL under various settings [Behzadan and Munir, 2017, Huang and Zhu, 2019, Ma et al., 2019, Zhang et al., 2020b, Sun et al., 2021, Rakhsha et al., 2020, 2021].

37th Conference on Neural Information Processing Systems (NeurIPS 2023).

On the other hand, except the ones that will be reviewed below, existing work on adversarial attacks on MARL is limited. In this paper, we aim to fill in this gap and systematically investigate the impact of adversarial attacks on online MARL. We consider a setting in which there is an attacker sits between the agents and the environment, and can monitor the states, the actions of the agents and the reward signals from the environment. The attacker is able to manipulate the feedback or action of the agents. The objective of the MARL learner is to learn an equilibrium. The attacker's goal is to force the agents to learn a target policy or to maximize the cumulative rewards under some specific reward function chosen by the attacker, while minimizing the amount of the manipulation on feedback and action. Our contributions are follows.

1) We propose an adversarial attack model in which the attacker aims to force the agent to learn a policy selected by the attacker (will be called target policy in the sequel) or to maximize the cumulative rewards under some specific reward function chosen by the attacker. We use loss and cost functions to evaluate the effectiveness of the adversarial attack on MARL agents. The cost is the cumulative sum of the action manipulations and the reward manipulations. If the attacker aims to force the agents to learn a target policy, the loss is the cumulative number of times when the agent does not follow the target policy. Otherwise, the loss is the regret to the policy that maximizes the attacker's rewards. It is clearly of interest to minimize both the loss and cost.

2) We study the attack problem in three different settings: the white-box, the gray-box and the black-box settings. In the white-box setting, the attacker has full information of the underlying environment. In the gray-box setting, the attacker has no prior information about the underlying environment and the agents' algorithm, but knows the target policy that maximizes its cumulative rewards. In the black-box setting, the target policy is also unknown for the attacker.

3) We show that the effectiveness of action poisoning only attacks and reward poisoning only attacks is limited. Even in the white-box setting, we show that there exist some MGs under which no action poisoning only Markov attack strategy or reward poisoning only Markov attack strategy can be efficient and successful. At the same time, we provide some sufficient conditions under which the action poisoning only attacks or the reward poisoning only attacks can efficiently attack MARL algorithms. Under such conditions, we introduce an efficient action poisoning attack strategy and an efficient reward poisoning attack strategy, and analyze their cost and loss.

4) We introduce a mixed attack strategy in the gray-box setting and an approximate mixed attack strategy in the black-box setting. We show that the mixed attack strategy can force any sub-linear-regret MARL agents to choose actions according to the target policy specified by the attacker with sub-linear cost and sub-linear loss. We further investigate the impact of the approximate mixed attack strategy attack on V-learning [Jin et al., 2021], a simple, efficient, decentralized algorithm for MARL.

## 1.1 Related works

**Attacks on Single Agent RL:** Adversarial attacks on single agent RL have been studied in various settings [Behzadan and Munir, 2017, Huang and Zhu, 2019, Ma et al., 2019, Zhang et al., 2020b, Sun et al., 2021, Rakhsha et al., 2020, 2021]. For example, [Behzadan and Munir, 2017, Zhang et al., 2020b, Rangi et al., 2022] study online reward poisoning attacks in which the attacker could manipulate the reward signal before the agent receives it. [Liu and Lai, 2021] studies online action poisoning attacks in which the attacker could manipulate the action signal before the environment receives it. [Rangi et al., 2022] studies the limitations of reward only manipulation or action only manipulation in single-agent RL.

**Attacks on MARL:** [Ma et al., 2022] considers a game redesign problem where the designer knows the full information of the game and can redesign the reward functions. The proposed redesign methods can incentivize players to take a specific target action profile frequently with a small cumulative design cost. [Gleave et al., 2020, Guo et al., 2021] study the poisoning attack on multi-agent reinforcement learners, assuming that the attacker controls one of the learners. [Wu et al., 2022] studies the reward poisoning attack on offline multi-agent reinforcement learners.

**Defense Against Attacks on RL:** There is also recent work on defending against adversarial attacks on RL [Banihashem et al., 2021, Zhang et al., 2021b, Lykouris et al., 2021, Chen et al., 2021, Wei et al., 2022, Wu et al., 2021]. These work focus on the single-agent RL setting where an adversary can corrupt the reward and state transition.

## 2 Problem setup

### 2.1 Definitions

To increase the readability of the paper, we first introduce some standard definitions related to MARL that will be used throughout of the paper. These definitions mostly follow those defined in [Jin et al., 2021]. We denote a tabular episodic MG with $m$ agents by a tuple $MG(\mathcal{S}, \{\mathcal{A}_i\}_{i=1}^m, H, P, \{R_i\}_{i=1}^m)$, where $\mathcal{S}$ is the state space with $|\mathcal{S}| = S$, $\mathcal{A}_i$ is the action space for the $i^{\text{th}}$ agent with $|\mathcal{A}_i| = A_i$, $H \in \mathbb{Z}^+$ is the number of steps in each episode. We let $\boldsymbol{a} := (a_1, \cdots, a_m)$ denote the joint action of all the $m$ agents and $\mathcal{A} := \mathcal{A}_i \times \cdots \times \mathcal{A}_m$ denote the joint action space. $P = \{P_h\}_{h \in [H]}$ is a collection of transition matrices. $P_h : \mathcal{S} \times \mathcal{A} \times \mathcal{S} \to [0, 1]$ is the probability transition function that maps state-action-state pair to a probability, $R_{i,h} : \mathcal{S} \times \mathcal{A} \to [0, 1]$ represents the reward function for the $i^{\text{th}}$ agent in the step $h$. In this paper, the probability transition functions and the reward functions can be different at different steps. We note that this MG model incorporates both cooperation and competition because the reward functions of different agents can be arbitrary.

**Interaction protocol:** The agents interact with the environment in a sequence of episodes. The total number of episodes is $K$. In each episode $k \in [K]$ of MG, the initial states $s_1$ is generated randomly by a distribution $P_0(\cdot)$. Initial states may be different between episodes. At each step $h \in [H]$ of an episode, each agent $i$ observes the state $s_h$ and chooses an action $a_{i,h}$ simultaneously. After receiving the action, the environment generates a random reward $r_{i,h} \in [0, 1]$ for each agent $i$ derived from a distribution with mean $R_{i,h}(s_h, \boldsymbol{a}_h)$, and transits to the next state $s_{h+1}$ drawn from the distribution $P_h(\cdot|s_h, \boldsymbol{a}_h)$. $P_h(\cdot|s, \boldsymbol{a})$ represents the probability distribution over states if joint action $\boldsymbol{a}$ is taken for state $s$. The agent stops interacting with environment after $H$ steps and starts another episode. At each time step, the agents may observe the actions played by other agents.

**Policy and value function:** A Markov policy takes actions only based on the current state. The policy $\pi_{i,h}$ of agent $i$ at step $h$ is expressed as a mappings $\pi_{i,h} : \mathcal{S} \to \Delta_{\mathcal{A}_i}$. $\pi_{i,h}(a_i|s)$ represents the probability of agent $i$ taking action $a_i$ in state $s$ under policy $\pi_i$ at step $h$. A deterministic policy is a policy that maps each state to a particular action. For notation convenience, for a deterministic policy $\pi_i$, we use $\pi_{i,h}(s)$ to denote the action $a_i$ which satisfies $\pi_{i,h}(a_i|s) = 1$. We denote the product policy of all the agents as $\pi := \pi_1 \times \cdots \times \pi_m$. We also denote $\pi_{-i} := \pi_1 \times \cdots \times \pi_{i-1} \times \pi_{i+1} \times \cdots \times \pi_m$ to be the product policy excluding agent $i$. If every agent follows a deterministic policy, the product policy of all the agents is also deterministic. We use $V_{i,h}^\pi : \mathcal{S} \to \mathbb{R}$ to denote the value function of agent $i$ at step $h$ under policy $\pi$ and define $V_{i,h}^\pi(s) := \mathbb{E}\left[\sum_{h'=h}^H r_{i,h'}|s_h = s, \pi\right]$. Given a policy $\pi$ and step $h$, the $i^{\text{th}}$ agent's $Q$-function $Q_{i,h}^\pi : \mathcal{S} \times \mathcal{A} \to \mathbb{R}$ of a state-action pair $(s, \boldsymbol{a})$ is defined as: $Q_{i,h}^\pi(s, \boldsymbol{a}) = \mathbb{E}\left[\sum_{h'=h}^H r_{i,h'}|s_h = s, \boldsymbol{a}_h = \boldsymbol{a}, \pi\right]$.

**Best response:** For any policy $\pi_{-i}$, there exists a best response of agent $i$, which is a policy that achieves the highest cumulative reward for itself if all other agents follow policy $\pi_{-i}$. We define the best response of agent $i$ towards policy $\pi_{-i}$ as $\mu^\dagger(\pi_{-i})$, which satisfies $\mu^\dagger(\pi_{-i}) := \arg\max_{\pi_i} V_{i,h}^{\pi_i \times \pi_{-i}}(s)$ for any state $s$ and any step $h$. We denote $\max_{\pi_i} V_{i,h}^{\pi_i \times \pi_{-i}}(s)$ as $V_{i,h}^{\dagger, \pi_{-i}}(s)$ for notation simplicity. By its definition, we know that the best response can always be achieved by a deterministic policy.

Nash Equilibrium (NE) is defined as a product policy where no agent can improve his own cumulative reward by unilaterally changing his strategy.

**Nash Equilibrium (NE) [Jin et al., 2021]:** A product policy $\pi$ is a NE if for all initial state $s$, $\max_{i \in [m]}(V_{i,1}^{\dagger, \pi_{-i}}(s) - V_{i,1}^\pi(s)) = 0$ holds. A product policy $\pi$ is an $\epsilon$-approximate Nash Equilibrium if for all initial state $s$, $\max_{i \in [m]}(V_{i,1}^{\dagger, \pi_{-i}}(s) - V_{i,1}^\pi(s)) \leq \epsilon$ holds.

**General correlated policy:** A general Markov correlated policy $\pi$ is a set of $H$ mappings $\pi := \{\pi_h : \Omega \times \mathcal{S} \to \Delta_{\mathcal{A}}\}_{h \in [H]}$. The first argument of $\pi_h$ is a random variable $\omega \in \Omega$ sampled from some underlying distributions. For any correlated policy $\pi = \{\pi_h\}_{h \in [H]}$ and any agent $i$, we can define a marginal policy $\pi_{-i}$ as a set of $H$ maps $\pi_i = \{\pi_{h,-i} : \Omega \times \mathcal{S} \to \Delta_{\mathcal{A}_{-i}}\}_{h \in [H]}$, where $\mathcal{A}_{-i} = \mathcal{A}_1 \times \cdots \times \mathcal{A}_{i-1} \times \mathcal{A}_{i+1} \times \cdots \times \mathcal{A}_m$. It is easy to verify that a deterministic joint policy is a product policy. The best response value of agent $i$ towards policy $\pi_{-i}$ as $\mu^\dagger(\pi_{-i})$, which satisfies $\mu^\dagger(\pi_{-i}) := \arg\max_{\pi_i} V_{i,h}^{\pi_i \times \pi_{-i}}(s)$ for any state $s$ and any step $h$.

**Coarse Correlated Equilibrium (CCE)[Jin et al., 2021]:** A correlated policy $\pi$ is an CCE if for all initial state $s$, $\max_{i\in[m]}(V_{i,1}^{\dagger,\pi_{-i}}(s) - V_{i,1}^{\pi}(s)) = 0$ holds. A correlated policy $\pi$ is an $\epsilon$-approximate CCE if for all initial state $s$, $\max_{i\in[m]}(V_{i,1}^{\dagger,\pi_{-i}}(s) - V_{i,1}^{\pi}(s)) \leq \epsilon$ holds.

**Strategy modification:** A strategy modification $\phi_i$ for agent $i$ is a set of mappings $\phi_i := \{(\mathcal{S} \times \mathcal{A})^{h-1} \times \mathcal{S} \times \mathcal{A}_i \to \mathcal{A}_i\}_{h\in[H]}$. For any policy $\pi_i$, the modified policy (denoted as $\phi_i \diamond \pi_i$) changes the action $\pi_{i,h}(\omega, s)$ under random sample $\omega$ and state $s$ to $\phi_i((s_1, \boldsymbol{a}_1, \ldots, s_h, a_{i,h}), \pi_{i,h}(\omega, s))$. For any joint policy $\pi$, we define the best strategy modification of agent $i$ as the maximizer of $\max_{\phi_i} V_{i,1}^{(\phi_i \diamond \pi_i) \odot \pi_{-i}}(s)$ for any initial state $s$.

**Correlated Equilibrium (CE)[Jin et al., 2021]:** A correlated policy $\pi$ is an CE if for all initial state $s$, $\max_{i\in[m]} \max_{\phi_i}(V_{i,1}^{(\phi_i \diamond \pi_i) \odot \pi_{-i}}(s) - V_{i,1}^{\pi}(s)) = 0$. A correlated policy $\pi$ is an $\epsilon$-approximate CE if for all initial state $s$, $\max_{i\in[m]} \max_{\phi_i}(V_{i,1}^{(\phi_i \diamond \pi_i) \odot \pi_{-i}}(s) - V_{i,1}^{\pi}(s)) \leq \epsilon$ holds.

In Markov games, it is known that an NE is an CE, and an CE is an CCE.

**Best-in-hindsight Regret:** Let $\pi^k$ denote the product policy deployed by the agents for each episode $k$. After $K$ episodes, the best-in-hindsight regret of agent $i$ is defined as $\text{Reg}_i(K, H) = \max_{\pi_i'} \sum_{k=1}^{K}[V_{i,1}^{\pi_i', \pi_{-i}^k}(s_1^k) - V_{i,1}^{\pi^k}(s_1^k)]$.

## 2.2 Poisoning attack setting

We are now ready to introduce the considered poisoning attack setting, in which there is an attacker sits between the agents and the environment. The attacker can monitor the states, the actions of the agents and the reward signals from the environment. Furthermore, the attacker can override actions and observations of agents. In particular, at each episode $k$ and step $h$, after each agent $i$ chooses an action $a_{i,h}^k$, the attacker may change it to another action $\widetilde{a}_{i,h}^k \in \mathcal{A}_i$. If the attacker does not override the actions, then $\widetilde{a}_{i,h}^k = a_i$. When the environment receives $\widetilde{\boldsymbol{a}}_h^k$, it generates random rewards $r_{i,h}^k$ with mean $R_{i,h}(s_h^k, \widetilde{\boldsymbol{a}}_h^k)$ for each agent $i$ and the next state $s_{h+1}^k$ is drawn from the distribution $P_h(\cdot|s_h^k, \widetilde{\boldsymbol{a}}_h^k)$. Before each agent $i$ receives the reward $r_{i,h}^k$, the attacker may change it to another reward $\widetilde{r}_{i,h}^k$. Agent $i$ receives the reward $\widetilde{r}_{i,h}^k$ and the next state $s_{h+1}^k$ from the environment. Note that agent $i$ does not know the attacker's manipulations and the presence of the attacker and hence will still view $\widetilde{r}_{i,h}^k$ as the reward and $s_{h+1}^k$ as the next state generated from state-action pair $(s_h^k, \boldsymbol{a}_h^k)$.

In this paper, we call an attack as *action poisoning only attack*, if the attacker only overrides the action but not the rewards. We call an attack as *reward poisoning only attack* if the attacker only overrides the rewards but not the actions. In addition, we call an attack as *mixed attack* if the attack can carry out both action poisoning and reward poisoning attacks simultaneously.

The goal of the MARL learners is to learn an equilibrium. On the other hand, the attacker's goal is to either force the agents to learn a target policy $\pi^{\dagger}$ of the attacker's choice or to force the agents to learn a policy that maximizes the cumulative rewards under a specific reward function $R_{\dagger,h} : \mathcal{S} \times \mathcal{A} \to (0, 1]$ chosen by the attacker. We note that this setup is very general. Different choices of $\pi^{\dagger}$ or $R_{\dagger,h}$ could lead to different objectives. For example, if the attacker aims to reduce the benefit of the agent $i$, the attacker's reward function $R_{\dagger,h}$ can be set to $1 - R_{i,h}$, or choose a target policy $\pi^{\dagger}$ that is detrimental to the agent $i$'s reward. If the attacker aims to maximize the total rewards of a subset of agents $\mathcal{C}$, the attacker's reward function $R_{\dagger,h}$ can be set to $\sum_{i\in\mathcal{C}} R_{i,h}$, or choose a target policy $\pi^{\dagger} = \arg\max \sum_{i\in\mathcal{C}} V_{i,1}^{\pi}(s_1)$ that maximizes the total rewards of agents in $\mathcal{C}$. We assume that the target policy $\pi^{\dagger}$ is deterministic and $R_{i,h}(s, \pi^{\dagger}(s)) > 0$. We measure the performance of the attack over $K$ episodes by the total attack cost and the attack loss. Set $\mathbb{1}(\cdot)$ as the indicator function. The attack cost over $K$ episodes is defined as $\text{Cost}(K, H) = \sum_{k=1}^{K} \sum_{h=1}^{H} \sum_{i=1}^{m} \left( \mathbb{1}(\widetilde{a}_{i,h}^k \neq a_{i,h}^k) + |\widetilde{r}_{i,h}^k - r_{i,h}^k| \right)$.

There are two different forms of attack loss based on the different goals of the attacker.

If the attacker's goal is to force the agents to learn a target policy $\pi^{\dagger}$, the attack loss over $K$ episodes is defined as $\text{Loss1}(K, H) = \sum_{k=1}^{K} \sum_{h=1}^{H} \sum_{i=1}^{m} \mathbb{1} \left( a_{i,h}^k \neq \pi_{i,h}^{\dagger}(s_{i,h}^k) \right)$.

If the attacker's goal is to force the agents to maximize the cumulative rewards under some specific reward function $R_{\dagger}$ chosen by the attacker, the attack loss over $K$ episodes is defined as $\text{Loss2}(K, H) =$

$\sum_{k=1}^{K}[V_{\dagger,1}^{\pi^*}(s_1^k) - V_{\dagger,1}^{\pi^k}(s_1^k)]$. Here, $V_{\dagger,1}^{\pi}(s)$ is the expected cumulative rewards in state $s$ based on the attacker's reward function $R_{\dagger}$ under product policy $\pi$ and $V_{\dagger,1}^{\pi^*}(s) = \max_{\pi} V_{\dagger,1}^{\pi}(s)$. $\pi^k$ denote the product policy deployed by the agents for each episode $k$. $\pi^*$ is the optimal policy that maximizes the attacker's cumulative rewards. We have $\text{Loss2}(K,H) \leq H * \text{Loss1}(K,H)$.

Denote the total number of steps as $T = KH$. In the proposed poisoning attack problem, we call an attack strategy *successful* if the attack loss of the strategy scales as $o(T)$. Furthermore, we call an attack strategy *efficient and successful* if both the attack cost and attack loss scale as $o(T)$.

The attacker aims to minimize both the attack cost and the attack loss, or minimize one of them subject to a constraint on the other. However, obtaining optimal solutions to these optimization problems is challenging. As the first step towards understanding the attack problem, we show the limitations of the action poisoning only or the reward poisoning only attacks and then propose a simple mixed attack strategy that is efficient and successful.

Depending on the capability of the attacker, we consider three settings: the white-box, the gray-box and the black-box settings. The table below summarizes the differences among these settings.

Table 1: Differences of the white/gray/black-box attackers

|  | white-box attacker | gray-box attacker | black-box attacker |
|---|---|---|---|
| MG | Has full information | No information | No information |
| $\pi^{\dagger}$ | Can be calculated if $R_{\dagger}$ given | Required and given | Not given |
| $R_{\dagger}$ | Not required if $\pi^{\dagger}$ given | Not required if $\pi^{\dagger}$ given | Required and given |
| Loss1 | Suitable by specify $\pi^{\dagger}$ | Suitable | Not suitable |
| Loss2 | Suitable if $R_{\dagger}$ given | Suitable if $R_{\dagger}$ given | Suitable |

## 3 White-box attack strategy and analysis

In this section, to obtain insights to the problem, we consider the white-box model, in which the attacker has full information of the underlying MG $(\mathcal{S}, \{\mathcal{A}_i\}_{i=1}^m, H, P, \{R_i\}_{i=1}^m)$. Even in the white-box attack model, we show that there exist some environments where the attacker's goal cannot be achieved by reward poisoning only attacks or action poisoning only attacks in Section 3.1. Then, in Section 3.2 and Section 3.3, we provide some sufficient conditions under which the action poisoning attacks alone or the reward poisoning attacks alone can efficiently attack MARL algorithms. Under such conditions, we then introduce an efficient action poisoning attack strategy and an efficient reward poisoning attack strategy.

### 3.1 The limitations of the action poisoning attacks and the reward poisoning attacks

As discussed in Section 2, the attacker aims to force the agents to either follow the target policy $\pi^{\dagger}$ or to maximize the cumulative rewards under attacker's reward function $R_{\dagger}$. In the white-box poisoning attack model, these two goals are equivalent as the optimal policy $\pi^*$ on the attacker's reward function $R_{\dagger}$ can be calculated by the Bellman optimality equations. To maximize the cumulative rewards under attacker's reward function $R_{\dagger}$ is equivalent to force the agents follow the policy $\pi^{\dagger} = \pi^*$.

Existing MARL algorithms [Liu et al., 2021, Jin et al., 2021] can learn an $\epsilon$-approximate {NE, CE, CCE} with $\widetilde{\mathcal{O}}(1/\epsilon^2)$ sample complexities. To force the MARL agents to follow the policy $\pi^{\dagger}$, the attacker first needs to attack the agents such that the target policy $\pi^{\dagger}$ is the unique NE in the observation of the agents. However, this alone is not enough to force the MARL agents to follow the policy $\pi^{\dagger}$. Any other distinct policy should not be an $\epsilon$-approximate CCE. The reason is that, if there exists an $\epsilon$-approximate CCE $\pi$ such that $\pi(\pi^{\dagger}(s)|s) = 0$ for any state $s$, the agents, using existing MARL algorithms, may learn and then follow $\pi$, which will lead the attack loss to be $\mathcal{O}(T) = \mathcal{O}(KH)$. Hence, we need to ensure that any $\epsilon$-approximate CCE stays in the neighborhood of the target policy. This requirement is equivalent to achieve the following objective: for all $s \in \mathcal{S}$, and policy $\pi$,

$$\max_{i \in [m]} (\widetilde{V}_{i,1}^{\dagger,\pi^{\dagger}_{-i}}(s) - \widetilde{V}_{i,1}^{\pi^{\dagger}}(s)) = 0;$$

if $\pi$ is a product policy and $\pi \neq \pi^{\dagger}$, then $\max_{i \in [m]} (\widetilde{V}_{i,1}^{\dagger,\pi_{-i}}(s) - \widetilde{V}_{i,1}^{\pi}(s)) > 0;$      (1)

if $\pi(\pi^{\dagger}(s')|s') = 0$ for all $s'$, then $\max_{i \in [m]} (\widetilde{V}_{i,1}^{\dagger,\pi_{-i}}(s) - \widetilde{V}_{i,1}^{\pi}(s)) > \epsilon,$

where $\widetilde{V}$ is the expected reward based on the post-attack environments.

We now investigate whether there exist efficient and successful attack strategies that use action poisoning alone or reward poisoning alone. We first show that the power of action poisoning attack alone is limited.

**Theorem 1** *There exists a target policy $\pi^\dagger$ and a MG $(\mathcal{S}, \{\mathcal{A}_i\}_{i=1}^m, H, P, \{R_i\}_{i=1}^m)$ such that no action poisoning Markov attack strategy alone can efficiently and successfully attack MARL agents by achieving the objective in (1).*

We now focus on strategies that use only reward poisoning. If the post-attack mean reward $\widetilde{R}$ is unbounded and the attacker can arbitrarily manipulate the rewards, there always exists an efficient and successful poisoning attack strategy. For example, the attacker can change the rewards of non-target actions to $-H$. However, such attacks can be easily detected, as the boundary of post-attack mean reward is distinct from the boundary of pre-attack mean reward. The following theorem shows that if the post-attack mean reward has the same boundary conditions as the pre-attack mean reward, the power of reward poisoning only attack is limited.

**Theorem 2** *If we limit the post-attack mean reward $\widetilde{R}$ to have the same boundary condition as that of the pre-attack mean reward R, i.e. $\widetilde{R} \in [0, 1]$, there exists a MG $(\mathcal{S}, \{\mathcal{A}_i\}_{i=1}^m, H, P, \{R_i\}_{i=1}^m)$ and a target policy $\pi^\dagger$ such that no reward poisoning Markov attack strategy alone can efficiently and successfully attack MARL agents by achieving the objective in (1).*

The proofs of Theorem 1 and Theorem 2 are provided in Appendix F. The main idea of the proofs is as follows. In successful poisoning attacks, the attack loss scales as $o(T)$ so that the agents will follow the target policy $\pi^\dagger$ in at least $T - o(T)$ times. To efficiently attack the MARL agents, the attacker should avoid to attack when the agents follow the target policy. Otherwise, the poisoning attack cost will grow linearly with $T$. The proofs of Theorem 1 and Theorem 2 proceed by constructing an MG and a target policy $\pi^\dagger$ where the expected rewards under $\pi^\dagger$ is always the worst for some agents if the attacker avoids to attack when the agents follow the target policy.

### 3.2 White-box action poisoning attacks

Even though Section 3.1 shows that there exists MG and target policy such that the action poisoning only attacks cannot be efficiently successful, here we show that it can be efficient and successful for a class of target policies. The following condition characterizes such class of target policies.

**Condition 1:** *For the underlying environment MG $(\mathcal{S}, \{\mathcal{A}_i\}_{i=1}^m, H, P, \{R_i\}_{i=1}^m)$, the attacker's target policy $\pi^\dagger$ satisfies that for any state $s$ and any step $h$, there exists an action $\boldsymbol{a}$ such that $V_{i,h}^{\pi^\dagger}(s) > Q_{i,h}^{\pi^\dagger}(s, \boldsymbol{a})$, for any agent $i$.*

Under Condition 1, we can find a worse policy $\pi^-$ by

$$\pi_h^-(s) = \arg\max_{\boldsymbol{a} \in \mathcal{A}} \min_{i \in [m]} \left( V_{i,h}^{\pi^\dagger}(s) - Q_{i,h}^{\pi^\dagger}(s, \boldsymbol{a}) \right) \ s.t. \forall i \in [m], V_{i,h}^{\pi^\dagger}(s) > Q_{i,h}^{\pi^\dagger}(s, \boldsymbol{a}). \quad (2)$$

Under this condition, we now introduce an effective white-box action attack strategies: $d$-portion attack. Specifically, at the step $h$ and state $s$, if all agents pick the target action, i.e., $\boldsymbol{a} = \pi_h^\dagger(s)$, the attacker does not attack, i.e. $\widetilde{\boldsymbol{a}} = \boldsymbol{a} = \pi_h^\dagger(s)$. If some agents pick a non-target action, i.e., $\boldsymbol{a} \neq \pi_h^\dagger(s)$, the $d$-portion attack sets $\widetilde{\boldsymbol{a}}$ as

$$\widetilde{\boldsymbol{a}} = \begin{cases} \pi_h^\dagger(s), \text{with probability } d_h(s, \boldsymbol{a})/m \\ \pi_h^-(s), \text{with probability } 1 - d_h(s, \boldsymbol{a})/m, \end{cases} \quad (3)$$

where $d_h(s, \boldsymbol{a}) = m/2 + \sum_{i=1}^m \mathbb{1}(a_i = \pi_{i,h}^\dagger(s))/2$.

**Theorem 3** *If the attacker follows the $d$-portion attack strategy on the MG agents, the best response of each agent $i$ towards the target policy $\pi_{-i}^\dagger$ is $\pi_i^\dagger$. The target policy $\pi^\dagger$ is an {NE, CE, CCE} from any agent's point of view. If every state $s \in \mathcal{S}$ is reachable at every step $h \in [H]$ under the target policy, $\pi^\dagger$ is the unique {NE, CE, CCE}.*

The detailed proof can be found in Appendix G.1. Theorem 3 shows that the target policy $\pi^\dagger$ is the unique {NE, CE, CCE} under the $d$-portion attack. Thus, if the agents follow an MARL algorithm that is able to learn an $\epsilon$-approximate {NE, CE, CCE}, the agents will learn a policy approximate to the target policy. We now discuss the high-level idea why the $d$-portion attack works. Under Condition 1, $\pi^-$ is worse than the target policy $\pi^\dagger$ at the step $H$ from every agent's point of view. Thus, under the $d$-portion attack, the target action strictly dominates any other action at the step $H$, and $\pi^\dagger$ is the unique {NE, CE, CCE} at the step $H$. From induction on $h = H, H - 1, \cdots, 1$, we can further prove that the $\pi^\dagger$ is the unique {NE, CE, CCE} at any step $h$. We define $\Delta_{i,h}^{\dagger -}(s) = Q_{i,h}^{\pi^\dagger}(s, \pi_h^\dagger(s)) - Q_{i,h}^{\pi^\dagger}(s, \pi_h^-(s))$ and the minimum gap $\Delta_{min} = \min_{h \in [H], s \in \mathcal{S}, i \in [m]} = \Delta_{i,h}^{\dagger -}(s)$. In addition, any other distinct policy is not an $\epsilon$-approximate CCE with different gap $\epsilon < \Delta_{min}/2$. We can derive upper bounds of the attack loss and the attack cost when attacking some special MARL algorithms.

**Theorem 4** *If the best-in-hindsight regret $Reg(K, H)$ of each agent's algorithm is bounded by a sub-linear bound $\mathcal{R}(T)$ for any MG in the absence of attack, and $\min_{s \in \mathcal{S}, i \in [m]} \Delta_{i,h}^{\dagger -}(s) \geq \sum_{h'=h+1}^{H} \max_{s \in \mathcal{S}, i \in [m]} \Delta_{i,h'}^{\dagger -}(s)$ holds for any $h \in [H]$, then $d$-portion attack will force the agents to follow the target policy with the attack loss and the attack cost bounded by*

$$\mathbb{E}[Loss1(K, H)] \leq 2m^2 \mathcal{R}(T)/\Delta_{min}, \ \mathbb{E}[Cost(K, H)] \leq 2m^3 \mathcal{R}(T)/\Delta_{min}. \tag{4}$$

### 3.3 White-box reward poisoning attacks

As stated in Theorem 2, the reward poisoning only attacks may fail, if we limit the post-attack mean reward $\widetilde{R}$ to satisfy the same boundary conditions as those of the pre-attack mean reward $R$, i.e. $\widetilde{R} \in [0, 1]$. However, similar to the case with action poisoning only attacks, the reward poisoning only attacks can be efficiently successful for a class of target policies. The following condition specifies such class of target policies.

**Condition 2:** *For the underlying environment MG $(\mathcal{S}, \{\mathcal{A}_i\}_{i=1}^m, H, P, \{R_i\}_{i=1}^m)$, there exists constant $\eta > 0$ such that for any state $s$, any step $h$, and any agent $i$, $(R_{i,h}(s, \pi^\dagger(s)) - \eta)/(H - h) \geq \Delta_R > 0$ where $\Delta_R = [\max_{s \times a \times h'} R_{i,h'}(s, a) - \min_{s \times a \times h'} R_{i,h'}(s, a)]$.*

We now introduce an effective white-box reward attack strategies: $\eta$-gap attack. Specifically, at the step $h$ and state $s$, if agents all pick the target action, i.e., $\boldsymbol{a} = \pi_h^\dagger(s)$, the attacker does not attack, i.e. $\widetilde{r}_{i,h} = r_{i,h}$ for each agent $i$. If agent $i$ picks a non-target action, i.e., $\boldsymbol{a} \neq \pi_h^\dagger(s)$, the $\eta$-gap attack sets $\widetilde{r}_{i,h} = R_{i,h}(s, \pi^\dagger(s)) - (\eta + (H - h)\Delta_R)\mathbb{1}(a_i \neq \pi_{i,h}^\dagger(s))$ for each agent $i$. From Condition 2, we have $\widetilde{r}_{i,h} \in [0, 1]$.

**Theorem 5** *If the attacker follows the $\eta$-gap attack strategy on the MG agents, the best response of each agent $i$ towards any policy $\pi_{-i}$ is $\pi_i^\dagger$. The target policy $\pi^\dagger$ is the {NE, CE, CCE} from any agent's point of view. If every state $s \in \mathcal{S}$ is reachable at every step $h \in [H]$ under the target policy, $\pi^\dagger$ is the unique {NE, CE, CCE}.*

The detailed proof can be found in Appendix H.1. Theorem 5 shows that the target policy $\pi^\dagger$ is the unique {NE, CE, CCE} under the $\eta$-gap attack. Thus, if the agents follow an MARL algorithm that is able to learn an $\epsilon$-approximate {NE, CE, CCE}, the agents will learn a policy approximate to the target policy. Here, we discuss the high-level idea why the $\eta$-gap attack works. $\Delta_R$ is the difference between the upper bound and the lower bound of the mean rewards. Condition 2 implies that each action is close to other actions from every agent's point of view. Although we limit the post-attack mean reward $\widetilde{R}$ in $[0, 1]$, the target policy can still appear to be optimal by making small changing to the rewards. Under Condition 2 and the $\eta$-gap attacks, the target actions strictly dominates any other non-target actions by at least $\eta$ and any other distinct policy is not an $\epsilon$-approximate CCE with different gap $\epsilon < \eta$. Thus, $\pi^\dagger$ becomes the unique {NE, CE, CCE}. In addition, we can derive upper bounds of the attack loss and the attack cost when attacking MARL algorithms with sub-linear best-in-hindsight regret.

**Theorem 6** *If the best-in-hindsight regret $Reg(K, H)$ of each agent's algorithm is bounded by a sub-linear bound $\mathcal{R}(T)$ for any MG in the absence of attack, then $\eta$-gap attack will force the agents to follow the target policy with the attack loss and the attack cost bounded by*

$$\mathbb{E}[Loss1(k, H)] \leq m\mathcal{R}(T)/\eta, \ \mathbb{E}[Cost(K, H)] \leq m^2 \mathcal{R}(T)/\eta. \tag{5}$$

We note that proposed sufficient conditions (namely Condition 1 and Condition 2), under which the action poisoning only attacks or the reward poisoning only attacks can be efficient and successful, may be strict. They may not always hold in practice. This motivates us to investigate mixed attack strategy to be discussed in the sequel.

## 4  Gray-box attack strategy and analysis

In the gray-box attack setting, the attacker has no prior information about the underlying environment and the agents' algorithm, and it only observes samples generated when the agents interact with the environment. However, the attacker is given the target policy $\pi^\dagger$. Since the $\eta$-gap reward attack strategy and $d$-portion action attack strategy described in Section 3 for the white-box setting rely on the information of the underlying environment, these two attack strategies are not applicable in the gray-box setting. In addition, without the information of the underlying environment, the attacker cannot check whether the action poisoning attack alone or the reward poisoning attack alone can be efficiently successful. Building on insights obtained from the white-box attack strategies, we develop a mixed attack strategy for MG in the gray-box attack setting.

In the proposed mixed attack strategy, at the step $h$ and state $s$, if agent $i$ picks the target action, i.e., $a_{i,h} = \pi^\dagger_{i,h}(s)$, the attacker does not override the action and the reward, i.e. $\widetilde{a}_{i,h} = a_{i,h}$ and $\widetilde{r}_{i,h} = r_{i,h}$. If agent $i$ picks a non-target action, i.e., $a_{i,h} \neq \pi^\dagger_{i,h}(s)$, the attacker overrides its action $\widetilde{a}_{i,h} = \pi^\dagger_{i,h}(s)$ and then overrides the reward $\widetilde{r}_{i,h} = 0$.

**Theorem 7** *If the attacker follows the mixed attack strategy the best response of each agent $i$ towards any product policy $\pi_{-i}$ is $\pi^\dagger_i$. The optimal policy $\pi^\dagger$ is the unique {NE, CE, CCE}.*

The detailed proof can be found in Appendix I.1. Here, we discuss the high-level idea why the mixed attack works. Under the mixed attacks, the state transitions are the same over the different actions and the reward of the non-target actions is worse than the target action. Thus, in the post-attack environment, the target policy is better than any other policy from any agent's point of view, and any other distinct policy is not an $\epsilon$-approximate CCE with different gap $\epsilon < R_{min}$, where $R_{min} = \min_{h \in [H]} \min_{s \in \mathcal{S}} \min_{i \in [m]} R_{i,h}(s, \pi^\dagger_h(s))$. Thus, $\pi^\dagger$ is the unique {NE, CE, CCE}. In addition, we can derive upper bounds of the attack loss and the attack cost when attacking some special MARL algorithms.

**Theorem 8** *If the best-in-hindsight regret Reg$(K, H)$ of each agent's algorithm is bounded by a sub-linear bound $\mathcal{R}(T)$ for any MG in the absence of attacks, then the mixed attacks will force the agents to follow the target policy $\pi^\dagger$ with the attack loss and the attack cost bounded by*

$$\mathbb{E}[Loss1(K, H)] \leq m\mathcal{R}(T)/R_{min}, \ \mathbb{E}[Cost(K, H)] \leq 2m\mathcal{R}(T)/R_{min}. \tag{6}$$

## 5  Black-box attack strategy and analysis

In the black-box attack setting, the attacker has no prior information about the underlying environment and the agents' algorithm, and it only observes the samples generated when the agents interact with the environment. The attacker aims to maximize the cumulative rewards under some specific reward functions $R_\dagger$ chosen by the attacker. But unlike in the gray-box case, the corresponding target policy $\pi^\dagger$ is also unknown for the attacker. After each time step, the attacker will receive the attacker reward $r_\dagger$. Since the optimal (target) policy that maximizes the attacker's reward is unknown, the attacker needs to explore the environment to obtain the optimal policy. As the mixed attack strategy described in Section 4 for the gray-box setting relies on the knowledge of the target policy, it is not applicable in the black-box setting.

However, by collecting observations and evaluating the attacker's reward function and transition probabilities of the underlying environment, the attacker can perform an approximate mixed attack strategy. In particular, we propose an approximate mixed attack strategy that has two phases: the exploration phase and the attack phase. In the exploration phase, the attacker explores the environment to identify an approximate optimal policy, while in the attack phase, the attacker performs the mixed attack strategy and forces the agents to learn the approximate optimal policy. The total attack cost (loss) will be the sum of attack cost (loss) of these two phases.

In the exploration phase, the approximate mixed attack strategy uses an optimal-policy identification algorithm, which is summarized in Algorithm 1, listed in Appendix A. It will return an approximate

optimal policy $\pi^\dagger$. Note that $\pi^k$ denotes the product policy deployed by the agents for each episode $k$. $\overline{V}$ is the upper bound of $V^{\pi^*}$ and $\underline{V}$ is the lower bound of $V^{\pi^k}$. By minimizing $\overline{V} - \underline{V}$, Algorithm 1 finds an approximate optimal policy $\pi^\dagger$. Here, we assume that the reward on the approximate optimal policy $\pi^\dagger$ is positive, i.e. $R_{min} = \min_{h \in [H]} \min_{s \in \mathcal{S}} \min_{i \in [m]} R_{i,h}(s, \pi_h^\dagger(s)) > 0$. In the exploration phase, the attacker will override both the agents' actions and rewards.

After the exploration phase, the approximate mixed attack strategy performs the attack phase. The attacker will override both the agents' actions and rewards in this phase. At the step $h$ and state $s$, if agent $i$ picks the action $\pi_{i,h}^\dagger(s)$, the attacker does not override actions and rewards, i.e. $\widetilde{a}_{i,h} = a_{i,h}$ and $\widetilde{r}_{i,h} = r_{i,h}$. If agent $i$ picks action $a_{i,h} \neq \pi_{i,h}^\dagger(s)$, the attacker overrides the action $\widetilde{a}_{i,h} = a_{i,h}$ and then overrides the reward $\widetilde{r}_{i,h} = 0$. The attack strategy in the attack phase is same with the mixed attack strategy. From Theorem 7, in the attack phase, the best response of each agent $i$ towards product policy $\pi_{-i}^\dagger$ is $\pi_i^\dagger$ and $\pi^\dagger$ is the unique NE. Here, we discuss the high-level idea why the approximate mixed attack works. The attacker finds an approximate optimal policy $\pi^\dagger$ by Algorithm 1. If $\pi^*$ is close to $\pi^\dagger$ and the exploration phase is sub-linear time dependent, the performance of the approximate mixed attack strategy will be close to the mixed attack strategy. We build a confidence bound to show the value function difference between $\pi^*$ and $\pi^\dagger$ in the following lemma.

**Lemma 1** *If the attacker follows the Algorithm 1 in Appendix A on the agents, for any $\delta \in (0,1)$, with probability at least $1 - 5\delta$, the following bound holds:*

$$\mathbb{E}_{s_1 \sim P_0(\cdot)}[V_{\dagger,1}^{\pi^*}(s_1) - V_{\dagger,1}^{\pi^\dagger}(s_1)] \leq 2H^2 S \sqrt{2A \log(2SAH\tau/\delta)/\tau}. \tag{7}$$

We now investigate the impact of the approximate mixed attack strategy attack on V-learning [Jin et al., 2021], a simple, efficient, decentralized algorithm for MARL. The reader's convienience, we list V-learning in Appendix J.2.

**Theorem 9** *Suppose ADV_BANDIT_UPDATE of V-learning follows Algorithm 3 in Appendix J.2 and it chooses hyper-parameter $w_t = \alpha_t \left( \prod_{i=2}^t (1 - \alpha_i) \right)^{-1}$, $\gamma_t = \sqrt{\frac{H \log B}{Bt}}$ and $\alpha_t = \frac{H+1}{H+t}$. For given $K$ and any $\delta \in (0,1)$, let $\iota = \log(mHSAK/\delta)$. The attack loss and the attack cost of the approximate mixed attack strategy during these $K$ episodes are bounded by*

$$\mathbb{E}\left[Loss2(K,H)\right] \leq H\tau + \frac{40}{R_{min}} m\sqrt{H^9 ASK\iota} + 2H^2 SK\sqrt{2A\iota/\tau},$$

$$\mathbb{E}\left[Cost(K,H)\right] \leq 2mH\tau + \frac{80}{R_{min}}\sqrt{H^5 ASK\iota}. \tag{8}$$

*Let $\hat{\pi}$ be the executing output policy of V-learning, the attack loss of the executing output policy $\hat{\pi}$ is upper bounded by*

$$V_{\dagger,1}^{\pi^*}(s_1) - V_{\dagger,1}^{\hat{\pi}}(s_1) \leq \frac{20mS}{R_{min}} \sqrt{\frac{H^7 A\iota}{K}} + \frac{2\tau mH^2 S}{K} + 2H^2 S\sqrt{2A\iota/\tau}. \tag{9}$$

If we choose the stopping time of the exploration phase $\tau = K^{2/3}$, the attack loss and the attack cost of the approximate mixed attack strategy during these $K$ episodes are bounded by $\mathcal{O}(K^{2/3})$ and $V_{\dagger,1}^{\pi^*}(s_1) - V_{\dagger,1}^{\hat{\pi}}(s_1) \leq \mathcal{O}(K^{-1/3})$.

## 6  Numerical Results

In this section, we empirically compare the performance of the action poisoning only attack strategy ($d$-portion attack), the reward poisoning only attack strategy ($\eta$-gap attack) and the mixed attack strategy.

We consider a simple case of Markov game where $m = 2$, $H = 2$ and $|\mathcal{S}| = 3$. This Markov game is the example in Appendix F.2. The initial state is $s_1$ at $h = 1$ and the transition probabilities are:

$$P(s_2|s_1, a) = 0.9, P(s_3|s_1, a) = 0.1, \text{ if } a = (\text{Defect}, \text{Defect}),$$
$$P(s_2|s_1, a) = 0.1, P(s_3|s_1, a) = 0.9, \text{ if } a \neq (\text{Defect}, \text{Defect}). \tag{10}$$

Table 2: Reward matrices

| state $s_1$ | Cooperate | Defect | state $s_2$ | Cooperate | Defect | state $s_3$ | Cooperate | Defect |
|---|---|---|---|---|---|---|---|---|
| Cooperate | (1, 1) | (0.5, 0.5) | Cooperate | (1, 1) | (0.5, 0.5) | Cooperate | (1, 1) | (0.5, 0.5) |
| Defect | (0.5, 0.5) | (0.2, 0.2) | Defect | (0.5, 0.5) | (0.1, 0.1) | Defect | (0.5, 0.5) | (0.9, 0.9) |

The reward functions are expressed in the following Table 2.

We set the total number of episodes $K = 10^7$. We set two different target policies. For the first target policy, no action/reward poisoning Markov attack strategy alone can efficiently and successfully attack MARL agents. For the second target policy, the $d$-portion attack and the $\eta$-gap attack can efficiently and successfully attack MARL agents.

**Case 1.** The target policy is that the two agents both choose to defect at any state. As stated in Section 3 and Appendix 3.1, the Condition 1 and Condition 2 do not hold for this Markov game and target policy, and no action/reward poisoning Markov attack strategy alone can efficiently and successfully attack MARL agents.

In Figure 1, we illustrate the mixed attack strategy, the $d$-portion attack strategy and the $\eta$-gap attack strategy on V-learning agents for the proposed MG. The $x$-axis represents the episode $k$ in the MG. The $y$-axis represents the cumulative attack cost and attack loss that change over time steps. The results show that, the attack cost and attack loss of the mixed attack strategy sublinearly scale as $T$, but the attack cost and attack loss of the $d$-portion attack strategy and the $\eta$-gap attack strategy linearly scale as $T$, which is consistent with our analysis.

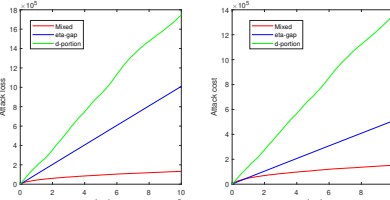

Figure 1: The attack loss (cost) on case 1.

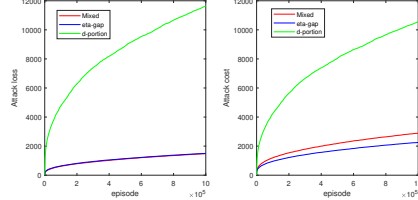

Figure 2: The attack loss (cost) on case 2.

**Case 2.** The target policy is that the two agents choose to cooperate at state $s_1$ and $s_2$ but to defect at state $s_3$. As stated in Section 3 and Appendix 3.1, the Condition 1 and Condition 2 hold for this Markov game and target policy. Thus, the $d$-portion attack strategy and the $\eta$-gap attack strategy can efficiently and successfully attack MARL agents.

In Figure 2, we illustrate the mixed attack strategy, the $d$-portion attack strategy and the $\eta$-gap attack strategy on V-learning agents for the proposed MG. The results show that, the attack cost and attack loss of all three strategies sublinearly scale as $T$, which is consistent with our analysis. Additional numerical results that compare the performance of the mixed attack strategy and the approximate mixed attack strategy are provided in Appendix B.

## 7 Conclusion

In this paper, we have introduced an adversarial attack model on MARL. We have discussed the attack problem in three different settings: the white-box, the gray-box and the black-box settings. We have shown that the power of action poisoning only attacks and reward poisoning only attacks is limited. Even in the white-box setting, there exist some MGs, under which no action poisoning only attack strategy or reward poisoning only attack strategy can be efficient and successful. We have then characterized conditions when action poisoning only attacks or only reward poisoning only attacks can efficiently work. We have further introduced the mixed attack strategy in the gray-box setting that can efficiently attack any sub-linear-regret MARL agents. Finally, we have proposed the approximate mixed attack strategy in the black-box setting and shown its effectiveness on V-learning. This paper raises awareness of the trustworthiness of online multi-agent reinforcement learning. In the future, we will investigate the defense strategy to mitigate the effects of this attack.

# 8    Acknowledgement

This work was supported by the National Science Foundation under Grant CCF-22-32907.

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

# A    The exploration phase of the approximate mixed attack strategy

The exploration phase of the approximate mixed attack strategy uses an optimal-policy identification algorithm, which is summarized in Algorithm 1. It will return an approximate optimal policy $\pi^\dagger$, which is an approximate optimal policy.

---

**Algorithm 1:** Exploration phase for Markov games

---

**Input:** Stopping time $\tau$. Set $B(N) = (H\sqrt{S} + 1)\sqrt{\log(2AH\tau/\delta)/(2N)}$.

1: Initialize $\overline{Q}_{\dagger,h}(s, \boldsymbol{a}) = \overline{V}_{\dagger,h}(s, \boldsymbol{a}) = H, \underline{Q}_{\dagger,h}(s, \boldsymbol{a}) = \underline{V}_{\dagger,h}(s, \boldsymbol{a}) = 0,$

$\overline{V}_{\dagger,H+1} = \underline{V}_{\dagger,H+1} = \mathbf{0}, \Delta = \infty, N_0(s) = N_h(s, \boldsymbol{a}) = N_h(s, \boldsymbol{a}, s') = 0$ and $\hat{R}_{\dagger,h}(s, \boldsymbol{a}) = 0$
   for any $(s, s', \boldsymbol{a}, i, h)$.

2: **for** episode $k = 1, \ldots, \tau$ **do**

3:    **for** step $h = H, \ldots, 1$ **do**

4:       **for** each $(s, \boldsymbol{a}) \in \mathcal{S} \times \mathcal{A}$ with $N_h(s, \boldsymbol{a}) > 0$ **do**

5:          Update $\overline{Q}_{\dagger,h}(s, \boldsymbol{a}) = \min\{\hat{R}_{\dagger,h} + \hat{\mathbb{P}}_h \overline{V}_{\dagger,h+1}(s, \boldsymbol{a}) + B(N_h(s, \boldsymbol{a})), H\}$ and

          $\underline{Q}_{\dagger,h}(s, \boldsymbol{a}) = \max\{\hat{R}_{\dagger,h} + \hat{\mathbb{P}}_h \underline{V}_{\dagger,h+1}(s, \boldsymbol{a}) - B(N_h(s, \boldsymbol{a})), 0\}.$

6:       **end for**

7:       **for** each $s \in \mathcal{S}$ with $N_h(s, \boldsymbol{a}) > 0$ **do**

8:          Update $\pi_h(s) = \max_{\boldsymbol{a} \in \mathcal{A}} \overline{Q}_{\dagger,h}(s, \boldsymbol{a}).$

9:          Update $\overline{V}_{\dagger,h}(s, \boldsymbol{a}) = \overline{Q}_{\dagger,h}(s, \pi_h(s))$ and $\underline{V}_{\dagger,h}(s, \boldsymbol{a}) = \underline{Q}_{\dagger,h}(s, \pi_h(s)).$

10:      **end for**

11:   **end for**

12:   **if** $\mathbb{E}_{s \sim \hat{\mathbb{P}}_0(\cdot)}(\overline{V}_{\dagger,1}(s) - \underline{V}_{\dagger,1}(s)) + H\sqrt{\frac{S\log(2\tau/\delta)}{2k}} \leq \Delta$ **then**

13:      $\Delta = \mathbb{E}_{s \sim \hat{\mathbb{P}}_0(\cdot)}(\overline{V}_{\dagger,1}(s) - \underline{V}_{\dagger,1}(s)) + H\sqrt{\frac{S\log(2\tau/\delta)}{2k}}$ and $\pi^\dagger = \pi.$

14:   **end if**

15:   **for** step $h = 1, \ldots, H$ **do**

16:      Attacker overrides each agent's action by changing $a_{i,h}$ to $\widetilde{a}_{i,h}$, where $\widetilde{\boldsymbol{a}}_h = \pi_h(s_h).$

17:      The environment returns the reward $r_{i,h}$ and the next state $s_{h+1}$ according to action $\widetilde{\boldsymbol{a}}_h$.
         The attacker receive its reward $r_{\dagger,h}$.

18:      Attacker overrides each agent's reward by changing $r_{i,h}$ to $\widetilde{r}_{i,h} = 1.$

19:      Add 1 to $N_h(s_h, \widetilde{\boldsymbol{a}}_h)$ and $N_h(s_h, \widetilde{\boldsymbol{a}}_h, s_{h+1})$. $\hat{\mathbb{P}}_h(\cdot|s_h, \widetilde{\boldsymbol{a}}_h) = N_h(s_h, \widetilde{\boldsymbol{a}}_h, \cdot)/N_h(s_h, \widetilde{\boldsymbol{a}}_h)$

20:      Update $\hat{R}_{\dagger,h}(s_h, \widetilde{\boldsymbol{a}}_h) = \hat{R}_{\dagger,h}(s_h, \widetilde{\boldsymbol{a}}_h) + (r_{\dagger,t} - \hat{R}_{\dagger,h}(s_h, \widetilde{\boldsymbol{a}}_h)/N_h(s_h, \widetilde{\boldsymbol{a}}_h).$

21:   **end for**

22:   Update $N_0(s_1) = N_0(s_1) + 1$ and $\hat{\mathbb{P}}_0(\cdot) = N_0(\cdot)/k.$

23: **end for**

24: Return $\pi^\dagger$.

---

# B    Additional numerical results

In this section, we empirically compare the performance of the mixed attack strategy and the approximate mixed attack strategy. We consider a multi-agent system with three recycling robots. In this scenario, a mobile robot with a rechargeable battery and a solar battery collects empty soda cans in a city. The number of agents is 3, i.e. $m = 3$. Each robot has two different energy levels, high energy level and low energy level, resulting in 8 states in total, i.e. $S = 8$.

Each robot can choose a conservative action or an aggressive action, so $A_i = 2$ and $A = 8$. At the high energy level, the conservative action is to wait in some place to save energy and then the mean reward is $0.4$. At the high energy level, the aggressive action is to search for cans. All the robots that choose to search will get a basic reward $0.2$ and equally share an additionally mean reward $0.9$. For example, if all robots choose to search at a step, the mean reward of each robot is $0.5$. At the low energy level, the conservative action is to return to change the battery and find the cans on the way. In this state and action, the robot only gets a low mean reward $0.2$. At the low energy level, the

conservative action is to wait in some place to save energy and then the mean reward is $0.3$. We use Gaussian distribution to randomize the reward signal.

We set the total number of steps $H = 6$. At the step $h \leq 3$, it is the daytime and the robot who chooses to search will change to the low energy level with low probability $0.3$. At the step $h \geq 4$, it is the night and the robot who chooses to search will change to the low energy level with high probability $0.7$. The energy level transition probabilities are stated in Figure 3 and Figure 4. 'H' represents the high energy level. 'L' represents the low energy level. 'C' represents the conservative action. 'A' represents the aggressive action.

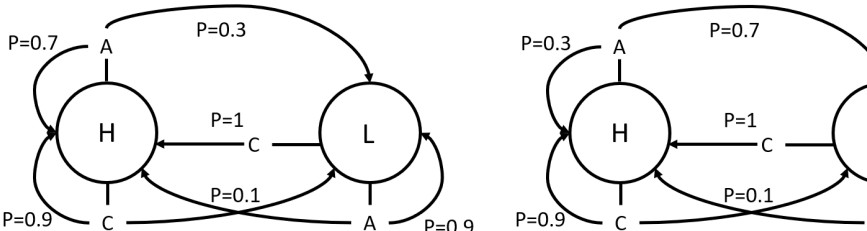

Figure 3: Energy level transitions at $h \leq 3$.      Figure 4: Energy level transitions at $h \geq 4$.

We consider two different attack goals: (1) maximize the first robot's rewards; (2) minimize the the second robot's and the third robot's rewards. For the gray box case, we provide the target policy that maximizes the first robot's rewards or minimizes the the second robot's and the third robot's rewards. For the black box case, we set $R_{\dagger,h} = R_{1,h}$ to maximize the first robot's rewards and set $R_{\dagger,h} = 1 - R_{2,h}/2 - R_{3,h}/2$ to minimize the second robot's and the third robot's rewards.

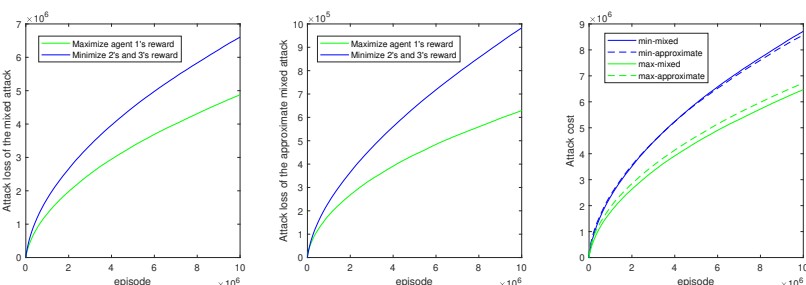

Figure 5: The cumulative attack loss and cost of the mixed attack and the approximate mixed attack.

We set the total number of episodes $K = 10^7$. In Figure 5, we illustrate the mixed attack strategy and approximate-mixed attack strategy on V-learning agents for the proposed MG. The $x$-axis represents the episode $k$ in the MG. The $y$-axis represents the cumulative attack cost and attack loss that change over time steps. The results show that, the attack cost and attack loss of the mixed attack strategy and approximate-mixed attack strategy sublinearly scale as $T$, which is consistent with our analysis. Furthermore, Figure 5 shows that the performance of the approximate-mixed attack strategy nearly match that of the mixed attack strategy. This illustrates that the proposed approximate-mixed attack strategy is very effective in the black-box scenario.

## C    Related works

Due to the page limit of the main paper, we do not provide a comprehensive comparison with prior research on adversarial attacks. We add the following discussion to Appendix.

Among the existing works on attacks in single-agent RL, the most related paper is [Rangi et al., 2022], which studies the limitations of reward only manipulation or action only manipulation in single-agent RL and proposed an attack strategy combining reward and action manipulation.

There are multiple differences between our work and [Rangi et al., 2022]. First, the MARL is modeled as a Markov game, but the single-agent RL is modeled as a MDP. In Markov game, each agent's

action will impact other agents' rewards. Second, the learning object of single-agent RL and MARL is different. The single-agent RL algorithms learn the optimal policy, but MARL algorithms learn the equilibrium. Since the attacks on one agent will impact all other agents and the equilibrium is considered as the agents' learning object, we have to develop techniques to carefully analyze the impact of attacks and the bound of the attack cost.

Here, we discuss the related work on the adversarial attacks on single MARL. [Ma et al., 2019] studies reward poisoning attack against batch RL in which the attacker is able to gather and modify the collected batch data. [Rakhsha et al., 2020] proposes a white-box environment poisoning model in which the attacker could manipulate the original MDP to a poisoned MDP. [Behzadan and Munir, 2017, Zhang et al., 2020b, Rangi et al., 2022] study online white-box reward poisoning attacks in which the attacker could manipulate the reward signal before the agent receives it. [Sun et al., 2021] proposes a practical black-box poisoning algorithm called VA2C-P. Their empirical results show that VA2C-P works for deep policy gradient RL agents without any prior knowledge of the environment. [Rakhsha et al., 2021] develops a black-box reward poisoning attack strategy called U2, that can provably attack any efficient RL algorithms. [Xu et al., 2021] investigates training-time attacks on RL agents and the introduced attacker can manipulate the environment.

Here, we discuss the related work on the adversarial attacks on MARL. [Ma et al., 2022] considers a game redesign problem where the designer knows the full information of the game and can redesign the reward functions. The proposed redesign methods can incentivize players to take a specific target action profile frequently with a small cumulative design cost. Ma's work considered the norm-form game but we considered the Markov game. The norm-form game is a simple case of the Markov game with horizon $H = 1$. [Gleave et al., 2020, Guo et al., 2021] study the poisoning attack on multi agent reinforcement learners, assuming that the attacker controls one of the learners. In our work, the attacker is not one of the learners, but an external unit out of the original Markov game. The attacker can poisoning the reward/action of all learners at the same time so that can fool the learners to learn a specific policy. [Wu et al., 2022] studies the reward poisoning attack on offline multi-agent reinforcement learners. The attacker can poisoning the reward of the agents. We considered the online MARL. In offline MARL, the attacker can estimate the underline Markov game from the offline datasets. In online MARL, the attacker may not have the knowledge (reward/transition functions) of the Markov game. [Wang et al., 2021] studies the backdoor attack in two-player competitive RL systems. The trigger is the action of another agent in the environment. They propose a unified method to design fast-failing agents which will fast fail when trigger occurred. [Liu et al., 2022] studies the controllable attack by constraining the state distribution shift caused by the adversarial policy and offering a more controllable attack scheme. [Chen et al., 2023] considers a situation that fraction of agents are adversarial and can report arbitrary fake information. They design two Byzantine-robust distributed value iteration algorithms that can identify a near-optimal policy with near-optimal sample complexity. [Mohammadi et al., 2023] studies targeted poisoning attacks in a two-agent setting where an attacker implicitly poisons the effective environment of one of the agents by modifying the policy of its peer.

# D Discussion

Due to the page limit of the main paper, we put the discussions regard the attack detection, the computational cost, the scalability of the attack strategies in Appendix.

**Attack detection**    We did not consider the attack detection in our problem, but the attack detection problem is also important. In this paper, we assumed that the agents do not know the existence of the attacker. Under this assumption, if the agents have no prior information of the MG, the proposed white and gray box attack is hard to be detected. As we consider the Markov attack strategy in this paper, the post-attack environment under the Markov attack strategy is still a Markov game. Without reference or prior information of the MG, the agents can not figure out whether the environment they observe is a post-attack environment or an attack-free environment. The proposed black attack may be detected, as the transition probabilities of the post-attack environment change over time. The goal of our paper is to understand and identify the impacts of different adversarial attacks. We hope our work can inspire follow-up work that can detect and mitigate such attacks so that RL models can be used in safety-critical applications. It is an important future direction for us to pursue.

**Computational cost**   For the proposed black-box attack strategy (the approximation mixed attack), the computational cost is $O(S^2 AH\tau + mKH)$. The proposed algorithm will compute the $Q$-values for each visited action-state pair at every steps and every episodes in the exploration phase. The computation of $Q$-value costs $O(S)$. Thus, the total computational cost in the exploration phase is $O(S^2 AH\tau)$. In the attack phase, the attacker only need to change the action and the reward for each agent so that the computational cost in the attack phase is $O(mKH)$.

**Scalability**   The gray-box attack strategies can be directly used in large-scale environments, even in some high-dimensional continuous environment. However, in the continuous space, the attacker does not change the non-target action to 0 but to $r_{i,h} * e^{c\|a_{i,h} - a_{i,h}^\dagger\|}$, in order to avoiding sparse reward. The ideas of the black-box attack strategies still work. However, the exploration phase should resort to some function approximation methods to efficiently explore an approximate target policy. Then the attack phase keeps the same as the gray-box attack.

# E   Notations

In this section, we introduce some notations that will be frequently used in appendixes.

The attack strategies in this paper are all Markov and only depend on the current state and actions. The post-attack reward function has the same form as the original reward function which is Markov and bounded in $[0, 1]$. Thus, the combination of the attacker and the environment $MG(\mathcal{S}, \{\mathcal{A}_i\}_{i=1}^m, H, P, \{R_i\}_{i=1}^m)$ can also be considered as a new environment $\widetilde{MG}(\mathcal{S}, \{\mathcal{A}_i\}_{i=1}^m, H, \widetilde{P}, \{\widetilde{R}_i\}_{i=1}^m)$, and the agents interact with the new environment. $\widetilde{R}_{i,h} : \mathcal{S} \times \mathcal{A} \to [0, 1]$ represents the post-attack reward function for the $i^{th}$ agent in the step $h$. The post-attack transition probabilities satisfy $\widetilde{P}_h(s'|s, \boldsymbol{a}) = \sum_{\boldsymbol{a}'} \mathbb{A}_h(\boldsymbol{a}'|s, \boldsymbol{a}) P_h(s'|s, \boldsymbol{a}')$.

We use $\widetilde{R}_i$, $\widetilde{N}_i$, $\widetilde{Q}_i$ and $\widetilde{V}_i$ to denote the mean rewards, counter, $Q$-values and value functions of the new post-attack environment that each agent $i$ observes. We use $N^k$, $V^k$ and $\pi^k$ to denote the counter, value functions, and policy maintained by the agents' algorithm at the beginning of the episode $k$.

For notation simplicity, we define two operators $\mathbb{P}$ and $\mathbb{D}$ as follows:

$$\begin{aligned}
\mathbb{P}_h[V](s, \boldsymbol{a}) &= \mathbb{E}_{s' \sim P_h(\cdot|s, \boldsymbol{a})} \left[ V(s') \right], \\
\mathbb{D}_\pi[Q](s) &= \mathbb{E}_{\boldsymbol{a} \sim \pi(\cdot|s)} \left[ Q(s, \boldsymbol{a}) \right].
\end{aligned} \tag{11}$$

Furthermore, we let $\mathbb{A}$ denote the action manipulation. $\mathbb{A} = \{\mathbb{A}_h\}_{h \in [H]}$ is a collection of action-manipulation matrices, so that $\mathbb{A}_h(\cdot|s, \boldsymbol{a})$ gives the probability distribution of the post-attack action if actions $\boldsymbol{a}$ are taken at state $s$ and step $h$. Using this notation, in the $d$-portion attack strategy, we have $\mathbb{A}_h(\pi_h^\dagger(s)|s, \boldsymbol{a}) = d_h(s, \boldsymbol{a})/m$, and $\mathbb{A}_h(\pi_h^-(s)|s, \boldsymbol{a}) = 1 - d_h(s, \boldsymbol{a})/m$.

# F   Proof of the insufficiency of action poisoning only attacks and reward poisoning only attacks

## F.1   Proof of Theorem 1

We consider a simple case of Markov game where $m = 2$, $H = 1$ and $|\mathcal{S}| = 1$. The reward function can be expressed in the matrix form in Table 3.

Table 3: Reward matrix

|           | Cooperate  | Defect     |
|-----------|------------|------------|
| Cooperate | (1, 1)     | (0.5, 0.5) |
| Defect    | (0.5, 0.5) | (0.1, 0.1) |

The target policy is that the two agents both choose to defect. In this MG, the two agents' rewards are the same under any action. As the action attacks only change the agent's action, the post-attack

Table 4: Post-attack reward matrix

|  | Cooperate | Defect |
|---|---|---|
| Cooperate | $(r_1, r_1)$ | $(r_2, r_2)$ |
| Defect | $(r_3, r_3)$ | $(r_4, r_4)$ |

rewards have the same property. The post-attack reward function can be expressed in the matrix form in Table 4.

To achieve the objective in (1), we first have $r_2 \leq r_4$ and $r_3 \leq r_4$, as the target policy should be an NE. Since the other distinct policy should not be an $\epsilon$-approximate CCE, we consider the other three pure-strategy policies and have

$$\begin{cases} r_1 > r_2 + \epsilon, \text{ or } r_4 > r_2 + \epsilon \\ r_1 > r_3 + \epsilon, \text{ or } r_4 > r_3 + \epsilon \;. \\ r_3 > r_1 + \epsilon, \text{ or } r_2 > r_1 + \epsilon \end{cases} \tag{12}$$

Note that $r_3 > r_1 + \epsilon$ and $r_1 > r_3 + \epsilon$ are contradictory and $r_2 > r_1 + \epsilon$ and $r_1 > r_2 + \epsilon$ are contradictory. We must have $r_4 > r_3 + \epsilon$ or $r_4 > r_2 + \epsilon$. As the action attacks will keep the same boundary of the rewards, $r_3 \geq 0.1$ and $r_2 \geq 0.1$. Then, $r_4 > 0.1 + \epsilon$.

Suppose there exists an action poisoning attack strategy that can successfully attack MARL agents. We have $\sum_{k=1}^{K} \sum_{h=1}^{H} \sum_{i=1}^{m} \mathbb{1}\left(a_{i,h}^k = \pi^\dagger(s_{i,h}^k)\right) = T - o(T) = \Omega(T)$, i.e. the attack loss scales on $o(T)$. To achieve the post-attack reward satisfy $r_4 > 0.1 + \epsilon$, the attacker needs to change the target action (Defect, Defect) to other actions with probability at least $\epsilon$, when the agents choose the target action. Then, we have $\sum_{k=1}^{K} \sum_{h=1}^{H} \sum_{i=1}^{m} \mathbb{E}(\mathbb{1}(\widetilde{a}_{i,h}^k \neq a_{i,h}^k)) = \Omega(\epsilon T)$. The expected attack cost is linearly dependent on $T$. Hence, there does not exist an action poisoning attack strategy that is both efficient and successful for this case.

### F.2 Proof of Theorem 2

We consider a simple case of Markov game where $m = 2$, $H = 2$ and $|\mathcal{S}| = 3$. The reward functions are expressed in the following Table 5.

Table 5: Reward matrix

| state $s_1$ | Cooperate | Defect |
|---|---|---|
| Cooperate | (1, 1) | (0.5, 0.5) |
| Defect | (0.5, 0.5) | (0.2, 0.2) |

| state $s_2$ | Cooperate | Defect |
|---|---|---|
| Cooperate | (1, 1) | (0.5, 0.5) |
| Defect | (0.5, 0.5) | (0.1, 0.1) |

| state $s_3$ | Cooperate | Defect |
|---|---|---|
| Cooperate | (1, 1) | (0.5, 0.5) |
| Defect | (0.5, 0.5) | (0.9, 0.9) |

The initial state is $s_1$ at $h = 1$ and the transition probabilities are:

$$\begin{aligned} P(s_2|s_1, a) = 0.9, P(s_3|s_1, a) = 0.1, \text{ if } a = \text{(Defect, Defect)}, \\ P(s_2|s_1, a) = 0.1, P(s_3|s_1, a) = 0.9, \text{ if } a \neq \text{(Defect, Defect)}. \end{aligned} \tag{13}$$

The target policy is that the two agents both choose to defect at any state. The post-attack reward function of the three states can be expressed in the matrix form in Table 6.

Table 6: Post-attack reward matrix

| state $s_1$ | Cooperate | Defect |
| --- | --- | --- |
| Cooperate | $(r_1, r_2)$ | $(r_3, r_4)$ |
| Defect | $(r_5, r_6)$ | $(r_7, r_8)$ |

| state $s_2$ | Cooperate | Defect |
| --- | --- | --- |
| Cooperate | ( - , - ) | ( - , - ) |
| Defect | ( - , - ) | $(r_9, r_{10})$ |

| state $s_3$ | Cooperate | Defect |
| --- | --- | --- |
| Cooperate | ( - , - ) | ( - , - ) |
| Defect | ( - , - ) | $(r_{11}, r_{12})$ |

We limit that the post-attack mean reward $\widetilde{R}$ has the same boundary condition with that of the pre-attack mean reward $R$, i.e. $\widetilde{R} \in [0, 1]$. Then, $0 \leq r_1, \ldots, r_{12} \leq 1$.

Suppose there exists a reward poisoning attack strategy that can successfully attack MARL agents, we have $\sum_{k=1}^{K} \sum_{h=1}^{H} \sum_{i=1}^{m} \mathbb{1}\left(a_{i,h}^k = \pi^\dagger(s_{i,h}^k)\right) = T - o(T) = \Omega(T)$, i.e. the attack loss scales on $o(T)$.

If $|r_9 - 0.1| > 0.1$, $|r_{10} - 0.1| > 0.1$, $|r_{11} - 0.9| > 0.1$, or $|r_{12} - 0.9| > 0.1$, we have the attack cost $\sum_{k=1}^{K} \sum_{h=1}^{H} \sum_{i=1}^{m} \mathbb{E}(|\widetilde{r}_{i,h}^k - r_{i,h}^k|) = \Omega(0.1 * K) = \Omega(T)$. Thus, $|r_9 - 0.1| \leq 0.1$, $|r_{10} - 0.1| \leq 0.1$, $|r_{11} - 0.9| \leq 0.1$ and $|r_{12} - 0.9| \leq 0.1$.

For the target policy, we have $\widetilde{V}_{i,1}^{\pi^\dagger}(s_1) = r_7 + 0.9 * r_9 + 0.1 * r_{11}$. For the policy $\pi'$ with $\pi_1'(s_1) = $ (Cooperate, Defect), $\pi_2'(s_2) = $ (Defect, Defect), $\pi_2'(s_3) = $ (Defect, Defect), we have $\widetilde{V}_{i,1}^{\pi'}(s_1) = r_3 + 0.1 * r_9 + 0.9 * r_{11}$.

To achieve the objective in (1), the attacker should let the target policy to be an NE. Thus, we have $\widetilde{V}_{i,1}^{\pi^\dagger}(s_1) \geq \widetilde{V}_{i,1}^{\pi'}(s_1)$ and then $r_7 + 0.9 * r_9 + 0.1 r_{11} \geq r_3 + 0.1 * r_9 + 0.9 * r_{11}$. As $|r_9 - 0.1| \leq 0.1$ and $|r_{11} - 0.9| \leq 0.1$, we have $r_7 \geq r_3 + 0.48$. From the boundary condition, we have $r_3 \geq 0$ and then $r_7 \geq 0.48$. The attack cost scales at least on $\Omega(0.28 * T)$ for a successful reward attack strategy.

In summary, there does not exist an reward poisoning attack strategy that is both efficient and successful for this case.

## G    Analysis of the $d$-portion Attack

### G.1    Proof of Theorem 3

We assume that the minimum gap exists and is positive, i.e. $\Delta_{min} > 0$. This positive gap provides an opportunity for efficient action poisoning attacks.

We assume that the agent does not know the attacker's manipulations and the presence of the attacker. The attacker's manipulations on actions are stationary. We can consider the combination of the attacker and the environment $\mathrm{MG}(\mathcal{S}, \{\mathcal{A}_i\}_{i=1}^m, H, P, \{R_i\}_{i=1}^m)$ as a new environment $\widetilde{\mathrm{MG}}(\mathcal{S}, \{\mathcal{A}_i\}_{i=1}^m, H, \widetilde{P}, \{\widetilde{R}_i\}_{i=1}^m)$, and the agents interact with the new environment. We define $\widetilde{Q}_i$ and $\widetilde{V}_i$ as the $Q$-values and value functions of the new environment $\widetilde{\mathrm{MG}}$ that each agent $i$ observes.

We first prove that $\pi^\dagger$ is an NE from every agent's point of view.

Condition 1 implies that $\pi^\dagger$ is not the worst policy from every agent's point of view, and there exists a policy $\pi^-$ that is worse than the target policy from every agent's point of view. Denote $\Delta_{i,h}^{\dagger-}(s) = Q_{i,h}^{\pi^\dagger}(s, \pi_h^\dagger(s)) - Q_{i,h}^{\pi^\dagger}(s, \pi_h^-(s))$. We define the minimum gap $\Delta_{min} = \min_{h \in [H], s \in \mathcal{S}, i \in [m]} = \Delta_{i,h}^{\dagger-}(s)$.

We set $\mathbb{P}_h V_{i,h+1}^{\pi}(s, \boldsymbol{a}) = \mathbb{E}_{s' \sim P_h(\cdot|s,\boldsymbol{a})} \left[ V_{i,h+1}^{\pi}(s') \right]$. From $d$-portion attack strategy, we have

$$\widetilde{Q}_{i,h}^{\pi}(s, \boldsymbol{a}) = \widetilde{R}_{i,h}(s, \boldsymbol{a}) + \frac{d_h(s, \boldsymbol{a})}{m} \mathbb{P}_h \widetilde{V}_{i,h+1}^{\pi}(s, \pi_h^{\dagger}(s)) + \left( 1 - \frac{d_h(s, \boldsymbol{a})}{m} \right) \mathbb{P}_h \widetilde{V}_{i,h+1}^{\pi}(s, \pi_h^{-}(s)), \tag{14}$$

and

$$\widetilde{R}_{i,h}(s, \boldsymbol{a}) = \frac{d_h(s, \boldsymbol{a})}{m} R_{i,h}(s, \pi_h^{\dagger}(s)) + \left( 1 - \frac{d_h(s, \boldsymbol{a})}{m} \right) R_{i,h}(s, \pi_h^{-}(s)). \tag{15}$$

Since the attacker does not attack when the agents follow the target policy, we have $\widetilde{V}_{i,h+1}^{\pi^{\dagger}}(s) = V_{i,h+1}^{\pi^{\dagger}}(s)$. Then,

$$\widetilde{Q}_{i,h}^{\pi^{\dagger}}(s, \boldsymbol{a}) = \frac{d_h(s, \boldsymbol{a})}{m} Q_{i,h}^{\pi^{\dagger}}(s, \pi_h^{\dagger}(s)) + \left( 1 - \frac{d_h(s, \boldsymbol{a})}{m} \right) Q_{i,h}^{\pi^{\dagger}}(s, \pi_h^{-}(s)). \tag{16}$$

If $a_i \neq \pi_{i,h}^{\dagger}(s)$, we have

$$\widetilde{Q}_{i,h}^{\pi^{\dagger}}(s, \pi_{i,h}^{\dagger}(s) \times \boldsymbol{a}_{-i}) - \widetilde{Q}_{i,h}^{\pi^{\dagger}}(s, \boldsymbol{a}) = \frac{1}{2m} \left( Q_{i,h}^{\pi^{\dagger}}(s, \pi_h^{\dagger}(s)) - Q_{i,h}^{\pi^{\dagger}}(s, \pi_h^{-}(s)) \right) \geq \frac{\Delta_{min}}{2m}. \tag{17}$$

We have that policy $\pi_i^{\dagger}$ is best-in-hindsight policy towards the target policy $\pi_{-i}^{\dagger}$ at step $h$ in the observation of each agent $i$, i.e. $\widetilde{V}_{i,h+1}^{\pi_i^{\dagger} \times \pi_{-i}^{\dagger}}(s) = \widetilde{V}_{i,h+1}^{\dagger, \pi_{-i}^{\dagger}}(s)$ for any agent $i$, any state $s$ and any policy $\pi_{-i}$.

Since the above argument works for any step $h \in [H]$, we have that the best response of each agent $i$ towards the target product policy $\pi_{-i}^{\dagger}$ is $\pi_i^{\dagger}$ and the target policy is an {NE, CE, CCE} under $d$-portion attack.

Now we prove that the target policy $\pi_i^{\dagger}$ is the unique {NE, CE, CCE}, when every state $s \in \mathcal{S}$ is reachable at every step $h \in [H]$ under the target policy.

If there exists an CCE $\pi'$ under $d$-portion attack, we have $\max_{i \in [m]}(\widetilde{V}_{i,1}^{\dagger, \pi'_{-i}}(s) - \widetilde{V}_{i,1}^{\pi'}(s)) = 0$ for any initial state $s$.

At the step $H$, $\widetilde{Q}_{i,H}^{\pi}(s, \boldsymbol{a}) = \widetilde{R}_{i,H}(s, \boldsymbol{a})$. Since $R_{i,H}(s, \pi_H^{\dagger}(s)) \geq R_{i,H}(s, \pi_H^{-}(s)) + \Delta_{min}$ with $\Delta_{min} > 0$, the policy $\pi_{i,H}^{\dagger}$ is the unique best response towards any policy $\pi_{-i,H}$, i.e. $\widetilde{V}_{i,H}^{\pi_{i,H}^{\dagger}, \pi_{-i,H}}(s) = \widetilde{V}_{i,H}^{\dagger, \pi_{-i,H}}(s)$ and $\widetilde{V}_{i,H}^{\pi_{i,H}^{\dagger}, \pi_{-i,H}}(s) > \widetilde{V}_{i,H}^{\pi_{i,H}, \pi_{-i,H}}(s)$ for any $\pi_{i,H}(\cdot|s) \neq \pi_{i,H}^{\dagger}(\cdot|s)$. Thus, we have $\pi_H'(s_H) = \pi_H^{\dagger}(s_H)$ for any state $s_H$ that is reachable at the time step $H$ under policy $\pi'$. We assume that every state $s \in \mathcal{S}$ is reachable at every step $h \in [H]$ under the target policy. Under $d$-portion attacks, the post-attack action $\widetilde{\boldsymbol{a}}_h = \pi_h^{\dagger}(s)$ with probability more than 0.5. Thus, every state $s \in \mathcal{S}$ is reachable at every step $h \in [H]$ under any policy $\pi$ and $d$-portion attacks.

Recall that for any $a_i \neq \pi_{i,h}^{\dagger}(s)$,

$$\widetilde{Q}_{i,h}^{\pi^{\dagger}}(s, \pi_{i,h}^{\dagger}(s) \times \boldsymbol{a}_{-i}) - \widetilde{Q}_{i,h}^{\pi^{\dagger}}(s, \boldsymbol{a}) \geq \Delta_{min}/2m. \tag{18}$$

Suppose $\{\pi_{h'}'\}_{h'=h+1}^{H} = \{\pi_{h'}^{\dagger}\}_{h'=h+1}^{H}$ for any states. If $\pi_{i,h}'(\cdot|s) \neq \pi_{i,h}^{\dagger}(\cdot|s)$ at a reachable state $s$, we have

$$\begin{aligned}
\widetilde{V}_{i,h}^{\pi'}(s) &= \mathbb{E}_{\boldsymbol{a} \sim \pi_h'(\cdot|s)} [\widetilde{Q}_{i,h}^{\pi'}(s, \boldsymbol{a})] \\
&= \mathbb{E}_{\boldsymbol{a} \sim \pi_h'(\cdot|s)} [\widetilde{Q}_{i,h}^{\pi^{\dagger}}(s, \boldsymbol{a})] \\
&\leq \mathbb{E}_{\boldsymbol{a}_{-i} \sim \pi_{-i,h}'(\cdot|s)} [\widetilde{Q}_{i,h}^{\pi^{\dagger}}(s, \pi_{i,h}^{\dagger}(s) \times \boldsymbol{a}_{-i})] - c\Delta_{min}/2m \\
&= \widetilde{V}_{i,h}^{\pi^{\dagger} \times \pi'_{-i}}(s) - c\Delta_{min}/m
\end{aligned} \tag{19}$$

with some constant $c > 0$. Then, $\pi'$ is not an CCE in such a case.

From induction on $h = H, H-1, \cdots, 1$, $\pi' = \pi^\dagger$ for any states. If every state $s \in \mathcal{S}$ is reachable at every step $h \in [H]$ under the target policy, $\pi^\dagger$ is the unique {NE, CE, CCE}.

## G.2 Proof of Theorem 4

Consider an arbitrary Markov policy $\pi$. From $d$-portion attack strategy, we have

$$
\begin{aligned}
\widetilde{Q}_{i,h}^\pi(s, \boldsymbol{a}) =& \frac{d_h(s, \boldsymbol{a})}{m} R_{i,h}(s, \pi_h^\dagger(s)) + \frac{m - d_h(s, \boldsymbol{a})}{m} R_{i,h}(s, \pi_h^-(s)) \\
& + \frac{d_h(s, \boldsymbol{a})}{m} \mathbb{P}_h \widetilde{V}_{i,h+1}^\pi(s, \pi_h^\dagger(s)) + \frac{m - d_h(s, \boldsymbol{a})}{m} \mathbb{P}_h \widetilde{V}_{i,h+1}^\pi(s, \pi_h^-(s)) \\
=& \frac{d_h(s, \boldsymbol{a}) - m}{m} \left( R_{i,h}(s, \pi_h^\dagger(s)) - R_{i,h}(s, \pi_h^-(s)) \right) \\
& + \frac{d_h(s, \boldsymbol{a}) - m}{m} \left( \mathbb{P}_h \widetilde{V}_{i,h+1}^\pi(s, \pi_h^\dagger(s)) - \mathbb{P}_h \widetilde{V}_{i,h+1}^\pi(s, \pi_h^-(s)) \right) \\
& + R_{i,h}(s, \pi_h^\dagger(s)) + \mathbb{P}_h \widetilde{V}_{i,h+1}^\pi(s, \pi_h^\dagger(s))
\end{aligned}
\tag{20}
$$

and

$$
\begin{aligned}
\widetilde{V}_{i,h}^\pi(s) =& \mathbb{D}_{\pi_h}[\widetilde{Q}_{i,h}^\pi](s) \\
=& \frac{\mathbb{D}_{\pi_h}[d](s) - m}{m} \left( R_{i,h}(s, \pi_h^\dagger(s)) - R_{i,h}(s, \pi_h^-(s)) \right) \\
& + \frac{\mathbb{D}_{\pi_h}[d](s) - m}{m} \left( \mathbb{P}_h \widetilde{V}_{i,h+1}^\pi(s, \pi_h^\dagger(s)) - \mathbb{P}_h \widetilde{V}_{i,h+1}^\pi(s, \pi_h^-(s)) \right) \\
& + R_{i,h}(s, \pi_h^\dagger(s)) + \mathbb{P}_h \widetilde{V}_{i,h+1}^\pi(s, \pi_h^\dagger(s)).
\end{aligned}
\tag{21}
$$

Now we bound the difference between $\widetilde{V}_{i,h}^\pi(s)$ and $\widetilde{V}_{i,h}^{\pi^\dagger}(s)$ for any policy $\pi$.

$$
\widetilde{V}_{i,h}^{\pi^\dagger}(s) - \widetilde{V}_{i,h}^\pi(s) = \underbrace{\widetilde{V}_{i,h}^{\pi^\dagger}(s) - \mathbb{D}_{\pi_h}[\widetilde{Q}_{i,h}^{\pi^\dagger}](s)}_{(a)} + \underbrace{\mathbb{D}_{\pi_h}[\widetilde{Q}_{i,h}^{\pi^\dagger}](s) - \widetilde{V}_{i,h}^\pi(s)}_{(b)}.
\tag{22}
$$

For term (a), from equations (20) and (21), we have

$$
\begin{aligned}
& \widetilde{V}_{i,h}^{\pi^\dagger}(s) - \mathbb{D}_{\pi_h}[\widetilde{Q}_{i,h}^{\pi^\dagger}](s) \\
=& \frac{m - \mathbb{D}_{\pi_h}[d](s)}{m} \left( R_{i,h}(s, \pi_h^\dagger(s)) - R_{i,h}(s, \pi_h^-(s)) \right) \\
& + \frac{m - \mathbb{D}_{\pi_h}[d](s)}{m} \left( \mathbb{P}_h \widetilde{V}_{i,h+1}^{\pi^\dagger}(s, \pi_h^\dagger(s)) - \mathbb{P}_h \widetilde{V}_{i,h+1}^{\pi^\dagger}(s, \pi_h^-(s)) \right).
\end{aligned}
\tag{23}
$$

Since the attacker does not attack when the agents follow the target policy, we have $\widetilde{V}_{i,h+1}^{\pi^\dagger}(s) = V_{i,h+1}^{\pi^\dagger}(s)$.

$$
\widetilde{V}_{i,h}^{\pi^\dagger}(s) - \mathbb{D}_{\pi_h}[\widetilde{Q}_{i,h}^{\pi^\dagger}](s) = \frac{m - \mathbb{D}_{\pi_h}[d](s)}{m} \left( Q_{i,h}^{\pi^\dagger}(s, \pi_h^\dagger(s)) - Q_{i,h}^{\pi^\dagger}(s, \pi_h^-(s)) \right).
\tag{24}
$$

Denote $\Delta_{i,h}^{\dagger-}(s) = Q_{i,h}^{\pi^\dagger}(s, \pi_h^\dagger(s)) - Q_{i,h}^{\pi^\dagger}(s, \pi_h^-(s))$. We have

$$
\widetilde{V}_{i,h}^{\pi^\dagger}(s) - \mathbb{D}_{\pi_h}[\widetilde{Q}_{i,h}^{\pi^\dagger}](s) = \frac{\Delta_{i,h}^{\dagger-}(s)}{2m} \mathbb{E}_{\boldsymbol{a} \sim \pi_h(\cdot|s)} \left[ \sum_{i=1}^m \mathbb{1}(a_i \neq \pi_{i,h}^\dagger(s)) \right].
\tag{25}
$$

For term (b), from equations (20) and (21), we have

$$
\begin{aligned}
&\mathbb{D}_{\pi_h}[\widetilde{Q}_{i,h}^{\pi^\dagger}](s) - \widetilde{V}_{i,h}^{\pi}(s)\\
=&\frac{\mathbb{D}_{\pi_h}[d](s)}{m}\mathbb{P}_h\widetilde{V}_{i,h+1}^{\pi^\dagger}(s,\pi_h^\dagger(s)) + \frac{m - \mathbb{D}_{\pi_h}[d](s)}{m}\mathbb{P}_h\widetilde{V}_{i,h+1}^{\pi^\dagger}(s,\pi_h^-(s))\\
&- \frac{\mathbb{D}_{\pi_h}[d](s)}{m}\mathbb{P}_h\widetilde{V}_{i,h+1}^{\pi}(s,\pi_h^\dagger(s)) - \frac{m - \mathbb{D}_{\pi_h}[d](s)}{m}\mathbb{P}_h\widetilde{V}_{i,h+1}^{\pi}(s,\pi_h^-(s))\\
=&\frac{\mathbb{D}_{\pi_h}[d](s)}{m}\mathbb{P}_h[\widetilde{V}_{i,h+1}^{\pi^\dagger} - \widetilde{V}_{i,h+1}^{\pi}](s,\pi_h^\dagger(s))\\
&+ \frac{m - \mathbb{D}_{\pi_h}[d](s)}{m}\mathbb{P}_h[\widetilde{V}_{i,h+1}^{\pi^\dagger} - \widetilde{V}_{i,h+1}^{\pi}](s,\pi_h^-(s))\\
=&\mathbb{E}_{s'\sim P_h(\cdot|s,\widetilde{\boldsymbol{a}}),\widetilde{\boldsymbol{a}}\sim\mathbb{A}_h(\cdot|s,\boldsymbol{a}),\boldsymbol{a}\sim\pi_h(\cdot|s)}[\widetilde{V}_{i,h+1}^{\pi^\dagger}(s') - \widetilde{V}_{i,h+1}^{\pi}(s')].
\end{aligned}
\tag{26}
$$

By combining terms (a) and (b), we have

$$
\begin{aligned}
&\widetilde{V}_{i,h}^{\pi^\dagger}(s_h) - \widetilde{V}_{i,h}^{\pi}(s_h)\\
=&\frac{\Delta_{i,h}^{\dagger-}(s_h)}{2m}\mathbb{E}_{\boldsymbol{a}\sim\pi_h(\cdot|s_h)}\left[\mathbb{1}(a_i \neq \pi_{i,h}^\dagger(s_h))\right]\\
&+ \mathbb{E}_{s_{h+1}\sim P_h(\cdot|s_h,\widetilde{\boldsymbol{a}}),\widetilde{\boldsymbol{a}}\sim\mathbb{A}_h(\cdot|s_h,\boldsymbol{a}),\boldsymbol{a}\sim\pi_h(\cdot|s_h)}[\widetilde{V}_{i,h+1}^{\pi^\dagger}(s_{h+1}) - \widetilde{V}_{i,h+1}^{\pi}(s_{h+1})]\\
=&\cdots = \mathbb{E}_{\pi,\mathbb{A},P}\left[\sum_{h'=h}^{H}\sum_{i=1}^{m}\mathbb{1}(a_{i,h'} \neq \pi_{i,h'}^\dagger(s_{h'}))\frac{\Delta_{i,h'}^{\dagger-}(s_{h'})}{2m}\right].
\end{aligned}
\tag{27}
$$

From the definition of the best-in-hindsight regret and (27), we have

$$
\begin{aligned}
\mathrm{Reg}_i(K,H) =&\max_{\pi_i'}\sum_{k=1}^{K}[\widetilde{V}_{i,1}^{\pi_i'\times\pi_{-i}^k}(s_1^k) - \widetilde{V}_{i,1}^{\pi^k}(s_1^k)]\\
\geq&\sum_{k=1}^{K}[\widetilde{V}_{i,1}^{\pi_i^\dagger\times\pi_{-i}^k}(s_1^k) - \widetilde{V}_{i,1}^{\pi^k}(s_1^k)].
\end{aligned}
\tag{28}
$$

Now, we bound $\sum_{i=1}^{m}[\widetilde{V}_{i,1}^{\pi_i^\dagger\times\pi_{-i}}(s_1) - \widetilde{V}_{i,1}^{\pi}(s_1)]$ for any policy $\pi$. We introduce some special strategy modifications $\{\phi_{i,h}^\dagger\}_{h=1}^{H}$. For any $h' \geq h$, we have $\phi_{i,h}^\dagger \diamond \pi_{i,h'}(s) = \pi_{i,h'}^\dagger(s)$ and for any $h' < h$, we have $\phi_{i,h}^\dagger \diamond \pi_{i,h'}(s) = \pi_{i,h'}(s)$. Thus,

$$
\begin{aligned}
&\sum_{i=1}^{m}[\widetilde{V}_{i,1}^{\pi_i^\dagger\times\pi_{-i}}(s_1) - \widetilde{V}_{i,1}^{\pi}(s_1)]\\
=&\sum_{h=1}^{H}\sum_{i=1}^{m}[\widetilde{V}_{i,1}^{\phi_{i,h}^\dagger\diamond\pi_i\times\pi_{-i}}(s_1) - \widetilde{V}_{i,1}^{\phi_{i,h+1}^\dagger\diamond\pi_i\times\pi_{-i}}(s_1)].
\end{aligned}
\tag{29}
$$

When $h = H$, we have

$$
\begin{aligned}
&\sum_{i=1}^{m}\left(\widetilde{V}_{i,1}^{\phi_{i,H}^\dagger\diamond\pi_i\times\pi_{-i}}(s_1) - \widetilde{V}_{i,1}^{\phi_{i,H+1}^\dagger\diamond\pi_i\times\pi_{-i}}(s_1)\right)\\
=&\mathbb{E}_{\pi,\mathbb{A},P}\left[\sum_{i=1}^{m}\left(\widetilde{V}_{i,H}^{\phi_{i,H}^\dagger\diamond\pi_i\times\pi_{-i}}(s_H) - \widetilde{V}_{i,H}^{\phi_{i,H+1}^\dagger\diamond\pi_i\times\pi_{-i}}(s_H)\right)\right]\\
=&\mathbb{E}_{\pi,\mathbb{A},P}\left[\sum_{i=1}^{m}\left(\widetilde{V}_{i,H}^{\pi_i^\dagger\times\pi_{-i}}(s_H) - \widetilde{V}_{i,H}^{\pi}(s_H)\right)\right]\\
=&\mathbb{E}_{\pi,\mathbb{A},P}\left[\sum_{i=1}^{m}\mathbb{1}(a_{i,H} \neq \pi_{i,H}^\dagger(s_H))\frac{\Delta_{i,H}^{\dagger-}(s_H)}{2m}\right].
\end{aligned}
\tag{30}
$$

For $h < H$, we have

$$
\sum_{i=1}^{m} \left( \widetilde{V}_{i,1}^{\phi_{i,h}^{\dagger} \diamond \pi_i \times \pi_{-i}}(s_1) - \widetilde{V}_{i,1}^{\phi_{i,h+1}^{\dagger} \diamond \pi_i \times \pi_{-i}}(s_1) \right)
$$

$$
= \mathbb{E}_{\pi, \mathbb{A}, P} \left[ \sum_{i=1}^{m} \left( \widetilde{V}_{i,h}^{\phi_{i,h}^{\dagger} \diamond \pi_i \times \pi_{-i}}(s_h) - \widetilde{V}_{i,h}^{\phi_{i,h+1}^{\dagger} \diamond \pi_i \times \pi_{-i}}(s_h) \right) \right]
$$

$$
= \mathbb{E}_{\pi, \mathbb{A}, P} \left[ \sum_{i=1}^{m} \left( \mathbb{D}_{\phi_{i,h}^{\dagger} \diamond \pi_{i,h} \times \pi_{-i,h}} - \mathbb{D}_{\phi_{i,h+1}^{\dagger} \diamond \pi_{i,h} \times \pi_{-i,h}} \right) \left[ \widetilde{Q}_{i,h}^{\phi_{i,h+1}^{\dagger} \diamond \pi_i \times \pi_{-i}} \right] (s_h) \right]
$$

$$
= \mathbb{E}_{\pi, \mathbb{A}, P} \left[ \sum_{i=1}^{m} \frac{1 - \pi_{i,h}\left(s_h, \pi_{i,h}^{\dagger}(s_h)\right)}{2m} (\mathbb{D}_{\pi^{\dagger}} - \mathbb{D}_{\pi^{-}}) \left[ R_{i,h} + \mathbb{P}_h \widetilde{V}_{i,h+1}^{\phi_{i,h+1}^{\dagger} \diamond \pi_i \times \pi_{-i}} \right] (s_h) \right]
\tag{31}
$$

where the second equation holds as $\phi_{i,h}^{\dagger} \diamond \pi_i \times \pi_{-i} = \phi_{i,h+1}^{\dagger} \diamond \pi_i \times \pi_{-i}$ at any time step $h' > h$ and the last equation holds from equation (20).

Note that $Q_{i,h}^{\pi^{\dagger}} = R_{i,h} + \mathbb{P}_h V_{i,h+1}^{\pi^{\dagger}} = R_{i,h} + \mathbb{P}_h \widetilde{V}_{i,h+1}^{\pi^{\dagger}}$. From equation (31), we have

$$
\sum_{i=1}^{m} \left( \widetilde{V}_{i,1}^{\phi_{i,h}^{\dagger} \diamond \pi_i \times \pi_{-i}}(s_1) - \widetilde{V}_{i,1}^{\phi_{i,h+1}^{\dagger} \diamond \pi_i \times \pi_{-i}}(s_1) \right)
$$

$$
= \underbrace{\mathbb{E}_{\pi, \mathbb{A}, P} \left[ \sum_{i=1}^{m} \frac{1 - \pi_{i,h}\left(s_h, \pi_{i,h}^{\dagger}(s_h)\right)}{2m} (\mathbb{D}_{\pi^{\dagger}} - \mathbb{D}_{\pi^{-}}) \left[ Q_{i,h}^{\pi^{\dagger}} \right] (s_h) \right]}_{①}
$$

$$
+ \underbrace{\mathbb{E}_{\pi, \mathbb{A}, P} \left[ \sum_{i=1}^{m} \frac{1 - \pi_{i,h}\left(s_h, \pi_{i,h}^{\dagger}(s_h)\right)}{2m} (\mathbb{D}_{\pi^{\dagger}} - \mathbb{D}_{\pi^{-}}) \left[ \mathbb{P}_h \widetilde{V}_{i,h+1}^{\phi_{i,h+1}^{\dagger} \diamond \pi_i \times \pi_{-i}} - \mathbb{P}_h \widetilde{V}_{i,h+1}^{\pi^{\dagger}} \right] (s_h) \right]}_{②} .
\tag{32}
$$

Denote $\Delta_{i,h}^{\dagger -}(s) = Q_{i,h}^{\pi^{\dagger}}(s, \pi_h^{\dagger}(s)) - Q_{i,h}^{\pi^{\dagger}}(s, \pi_h^{-}(s))$. Thus,

$$
① = \mathbb{E}_{\pi, \mathbb{A}, P} \left[ \sum_{i=1}^{m} \frac{1 - \pi_{i,h}\left(s_h, \pi_{i,h}^{\dagger}(s_h)\right)}{2m} \Delta_{i,h}^{\dagger -}(s_h) \right] .
\tag{33}
$$

Now, we bound item ②. If $(\mathbb{D}_{\pi^{\dagger}} - \mathbb{D}_{\pi^{-}}) \left[ \mathbb{P}_h \widetilde{V}_{i,h+1}^{\phi_{i,h+1}^{\dagger} \diamond \pi_i \times \pi_{-i}} - \mathbb{P}_h \widetilde{V}_{i,h+1}^{\pi^{\dagger}} \right] (s_h) \geq 0$,

$$
(\mathbb{D}_{\pi^{\dagger}} - \mathbb{D}_{\pi^{-}}) \left[ \mathbb{P}_h \widetilde{V}_{i,h+1}^{\phi_{i,h+1}^{\dagger} \diamond \pi_i \times \pi_{-i}} - \mathbb{P}_h \widetilde{V}_{i,h+1}^{\pi^{\dagger}} \right] (s_h)
$$

$$
\geq \frac{2 \mathbb{D}_{\pi_h}[d](s_h)}{m} \mathbb{D}_{\pi^{\dagger}} \mathbb{P}_h [\widetilde{V}_{i,h+1}^{\phi_{i,h+1}^{\dagger} \diamond \pi_i \times \pi_{-i}} - \widetilde{V}_{i,h+1}^{\pi^{\dagger}}](s_h)
$$

$$
+ \frac{2(m - \mathbb{D}_{\pi_h}[d](s_h))}{m} \mathbb{D}_{\pi^{-}} \mathbb{P}_h [\widetilde{V}_{i,h+1}^{\phi_{i,h+1}^{\dagger} \diamond \pi_i \times \pi_{-i}} - \widetilde{V}_{i,h+1}^{\pi^{\dagger}}](s_h)
$$

$$
= 2 \mathbb{E}_{\pi, \mathbb{A}, P} \left[ \widetilde{V}_{i,h+1}^{\phi_{i,h+1}^{\dagger} \diamond \pi_i \times \pi_{-i}}(s_{h+1}) - \widetilde{V}_{i,h+1}^{\pi^{\dagger}}(s_{h+1}) \right] ,
\tag{34}
$$

because the RHS of the inequality is smaller or equal to 0.

If $(\mathbb{D}_{\pi^\dagger} - \mathbb{D}_{\pi^-}) \left[ \mathbb{P}_h \widetilde{V}_{i,h+1}^{\phi_{i,h+1}^\dagger \diamond \pi_i \times \pi_{-i}} - \mathbb{P}_h \widetilde{V}_{i,h+1}^{\pi^\dagger} \right](s_h) \leq 0$,

$$
(\mathbb{D}_{\pi^\dagger} - \mathbb{D}_{\pi^-}) \left[ \mathbb{P}_h \widetilde{V}_{i,h+1}^{\phi_{i,h+1}^\dagger \diamond \pi_i \times \pi_{-i}} - \mathbb{P}_h \widetilde{V}_{i,h+1}^{\pi^\dagger} \right](s_h)
$$

$$
\geq \frac{2\mathbb{D}_{\pi_h}[d](s_h)}{m} (\mathbb{D}_{\pi^\dagger} - \mathbb{D}_{\pi^-}) \left[ \mathbb{P}_h \widetilde{V}_{i,h+1}^{\phi_{i,h+1}^\dagger \diamond \pi_i \times \pi_{-i}} - \mathbb{P}_h \widetilde{V}_{i,h+1}^{\pi^\dagger} \right](s_h)
$$

$$
\geq \frac{2\mathbb{D}_{\pi_h}[d](s_h)}{m} \mathbb{D}_{\pi^\dagger} \mathbb{P}_h [\widetilde{V}_{i,h+1}^{\phi_{i,h+1}^\dagger \diamond \pi_i \times \pi_{-i}} - \widetilde{V}_{i,h+1}^{\pi^\dagger}](s_h) \tag{35}
$$

$$
+ \frac{2(m - \mathbb{D}_{\pi_h}[d](s_h))}{m} \mathbb{D}_{\pi^-} \mathbb{P}_h [\widetilde{V}_{i,h+1}^{\phi_{i,h+1}^\dagger \diamond \pi_i \times \pi_{-i}} - \widetilde{V}_{i,h+1}^{\pi^\dagger}](s_h)
$$

$$
= 2\mathbb{E}_{\pi,\mathbb{A},P} \left[ \widetilde{V}_{i,h+1}^{\phi_{i,h+1}^\dagger \diamond \pi_i \times \pi_{-i}}(s_{h+1}) - \widetilde{V}_{i,h+1}^{\pi^\dagger}(s_{h+1}) \right].
$$

From (27), we have

$$
\widetilde{V}_{i,h+1}^{\pi^\dagger}(s_{h+1}) - \widetilde{V}_{i,h+1}^{\phi_{i,h+1}^\dagger \diamond \pi_i \times \pi_{-i}}(s_{h+1})
$$

$$
= \mathbb{E}_{\phi_{i,h+1}^\dagger \diamond \pi_i \times \pi_{-i}, \mathbb{A}, P} \left[ \sum_{h'=h+1}^{H} \sum_{i=1}^{m} \mathbb{1}(a_{i,h'} \neq \pi_{i,h'}^\dagger(s_{h'})) \frac{\Delta_{i,h'}^{\dagger-}(s_{h'})}{2m} \right] \tag{36}
$$

$$
\leq \sum_{h'=h+1}^{H} (m-1) \max_{s \in \mathcal{S}, i \in [m]} \frac{\Delta_{i,h'}^{\dagger-}(s)}{2m}
$$

$$
\leq \frac{(m-1)}{2m} \Delta_{i,h}^{\dagger-}(s_h),
$$

where the last inequality holds when $\min_{s \in \mathcal{S}, i \in [m]} \Delta_{i,h}^{\dagger-}(s) \geq \sum_{h'=h+1}^{H} \max_{s \in \mathcal{S}, i \in [m]} \Delta_{i,h'}^{\dagger-}(s)$. Combine the above inequalities, we have

$$
② \geq -\mathbb{E}_{\pi,\mathbb{A},P} \left[ \sum_{i=1}^{m} \frac{1 - \pi_{i,h}\left(s_h, \pi_{i,h}^\dagger(s_h)\right)}{2m} \frac{(m-1)}{m} \Delta_{i,h}^{\dagger-}(s_h) \right], \tag{37}
$$

and

$$
\sum_{i=1}^{m} \left( \widetilde{V}_{i,1}^{\phi_{i,h}^\dagger \diamond \pi_i \times \pi_{-i}}(s_1) - \widetilde{V}_{i,1}^{\phi_{i,h+1}^\dagger \diamond \pi_i \times \pi_{-i}}(s_1) \right)
$$

$$
= ① + ②
$$

$$
\geq \mathbb{E}_{\pi,\mathbb{A},P} \left[ \sum_{i=1}^{m} \frac{1 - \pi_{i,h}\left(s_h, \pi_{i,h}^\dagger(s_h)\right)}{2m^2} \Delta_{i,h}^{\dagger-}(s_h) \right] \tag{38}
$$

$$
= \mathbb{E}_{\pi,\mathbb{A},P} \left[ \sum_{i=1}^{m} \mathbb{1}(a_{i,h} \neq \pi_{i,h}^\dagger(s_h)) \frac{\Delta_{i,h}^{\dagger-}(s_h)}{2m^2} \right]
$$

$$
\geq \mathbb{E}_{\pi,\mathbb{A},P} \left[ \sum_{i=1}^{m} \mathbb{1}(a_{i,h} \neq \pi_{i,h}^\dagger(s_h)) \right] \frac{\Delta_{min}}{2m^2}.
$$

In summary,

$$
\mathrm{Reg}_i(K, H) \geq \sum_{h=1}^{H} \mathbb{E}_{\pi,\mathbb{A},P} \left[ \sum_{i=1}^{m} \mathbb{1}(a_{i,h} \neq \pi_{i,h}^\dagger(s_h)) \right] \frac{\Delta_{min}}{2m^2} = \mathbb{E}[\mathrm{Loss}1(K, H)] \frac{\Delta_{min}}{2m^2}. \tag{39}
$$

If the best-in-hindsight regret $\mathrm{Reg}(K, H)$ of each agent's algorithm is bounded by a sub-linear bound $\mathcal{R}(T)$, then the attack loss is bounded by $\mathbb{E}[\mathrm{Loss}1(K, H)] \leq 2m^2 \mathcal{R}(T)/\Delta_{min}$.

The $d$-portion attack strategy attacks all agents when any agent $i$ chooses an non-target action. We have

$$\text{Cost}(K,H) = \sum_{k=1}^{K}\sum_{h=1}^{H}\sum_{i=1}^{m}\left(\mathbb{1}(\widetilde{a}_{i,h}^{k} \neq a_{i,h}^{k}) + |\widetilde{r}_{i,h}^{k} - r_{i,h}^{k}|\right)$$

$$\leq \sum_{k=1}^{K}\sum_{h=1}^{H}\sum_{i=1}^{m}\mathbb{1}[\widetilde{a}_{i,h}^{k} \neq a_{i,h}^{k}]m. \tag{40}$$

Then, the attack cost is bounded by $m\mathbb{E}[\text{Loss1}(K,H)] \leq 2m^3\mathcal{R}(T)/\Delta_{min}$.

## H  Analysis of the $\eta$-gap attack

### H.1  Proof of Theorem 5

We assume that the agent does not know the attacker's manipulations and the presence of the attacker. The attacker's manipulations on rewards are stationary. We can consider the combination of the attacker and the environment $\text{MG}(\mathcal{S},\{\mathcal{A}_i\}_{i=1}^m, H, P, \{R_i\}_{i=1}^m)$ as a new environment $\widetilde{\text{MG}}(\mathcal{S},\{\mathcal{A}_i\}_{i=1}^m, H, P, \{\widetilde{R}_i\}_{i=1}^m)$, and the agents interact with the new environment. We define $\widetilde{Q}_i$ and $\widetilde{V}_i$ as the $Q$-values and value functions of the new environment $\widetilde{\text{MG}}$ that each agent $i$ observes.

We introduce some special strategy modifications $\{\phi_{i,h}^\dagger\}_{h=1}^H$. For any $h' \geq h$, we have $\phi_{i,h}^\dagger \diamond \pi_{i,h'}(s) = \pi_{i,h'}^\dagger(s)$ and for any $h' < h$, we have $\phi_{i,h}^\dagger \diamond \pi_{i,h'}(s) = \pi_{i,h'}(s)$. Thus,

$$\widetilde{V}_{i,1}^{\pi_i^\dagger \times \pi_{-i}}(s_1) - \widetilde{V}_{i,1}^{\pi}(s_1) = \sum_{h=1}^{H}[\widetilde{V}_{i,1}^{\phi_{i,h}^\dagger \diamond \pi_i \times \pi_{-i}}(s_1) - \widetilde{V}_{i,1}^{\phi_{i,h+1}^\dagger \diamond \pi_i \times \pi_{-i}}(s_1)]. \tag{41}$$

We have that for any policy $\pi$,

$$[\widetilde{V}_{i,1}^{\phi_{i,h}^\dagger \diamond \pi_i \times \pi_{-i}}(s_1) - \widetilde{V}_{i,1}^{\phi_{i,h+1}^\dagger \diamond \pi_i \times \pi_{-i}}(s_1)]$$

$$= \mathbb{E}_{\pi,\mathbb{A},P}\left[\left(\widetilde{V}_{i,h}^{\phi_{i,h}^\dagger \diamond \pi_i \times \pi_{-i}}(s_h) - \widetilde{V}_{i,h}^{\phi_{i,h+1}^\dagger \diamond \pi_i \times \pi_{-i}}(s_h)\right)\right]$$

$$= \mathbb{E}_{\pi,\mathbb{A},P}\left[\left(\mathbb{D}_{\phi_{i,h}^\dagger \diamond \pi_{i,h} \times \pi_{-i,h}} - \mathbb{D}_{\phi_{i,h+1}^\dagger \diamond \pi_{i,h} \times \pi_{-i,h}}\right)\left[\widetilde{Q}_{i,h}^{\phi_{i,h+1}^\dagger \diamond \pi_i \times \pi_{-i}}\right](s_h)\right] \tag{42}$$

$$= \mathbb{E}_{\pi,\mathbb{A},P}\left[\left(\mathbb{D}_{\pi_{i,h}^\dagger \times \pi_{-i,h}} - \mathbb{D}_{\pi_h}\right)\left[\widetilde{R}_{i,h} + \mathbb{P}_h\widetilde{V}_{i,h+1}^{\phi_{i,h+1}^\dagger \diamond \pi_i \times \pi_{-i}}\right](s_h)\right].$$

Since $\widetilde{R}_{i,h}(s,\boldsymbol{a}) = R_{i,h}(s,\pi^\dagger(s)) - (\eta + (H-h)\Delta_R)\mathbb{1}(a_i \neq \pi_{i,h}^\dagger(s))$ from $\eta$-gap attack strategy and $(H-h)\min_{s'\times a'\times h'}R_{i,h'}(s',a') < \mathbb{P}_h\widetilde{V}_{i,h+1}^\pi(s',a') \leq (H-h)\max_{s'\times a'\times h'}R_{i,h'}(s',a')$ for any $s$ and $a$, we have

$$[\widetilde{V}_{i,1}^{\phi_{i,h}^\dagger \diamond \pi_i \times \pi_{-i}}(s_1) - \widetilde{V}_{i,1}^{\phi_{i,h+1}^\dagger \diamond \pi_i \times \pi_{-i}}(s_1)]$$

$$= \mathbb{E}_{\pi,\mathbb{A},P}\left[\sum_{\boldsymbol{a}}\pi_h(\boldsymbol{a}|s_h)(\eta + (H-h)\Delta_R)\mathbb{1}(a_i \neq \pi_{i,h}^\dagger(s_h))\right]$$

$$+ \mathbb{E}_{\pi,\mathbb{A},P}\left[\sum_{\boldsymbol{a}}\pi_h(\boldsymbol{a}|s_h)\mathbb{1}(a_i \neq \pi_{i,h}^\dagger(s_h))\left(\mathbb{P}_h\widetilde{V}_{i,h+1}^{\phi_{i,h+1}^\dagger \diamond \pi_i \times \pi_{-i}}(s_h, \pi_{i,h}^\dagger(s_h) \times \boldsymbol{a}_{-i}) - \mathbb{P}_h\widetilde{V}_{i,h+1}^{\phi_{i,h+1}^\dagger \diamond \pi_i \times \pi_{-i}}(s_h, \boldsymbol{a})\right)\right]$$

$$\geq \mathbb{E}_{\pi,\mathbb{A},P}\left[\sum_{\boldsymbol{a}}\pi_h(\boldsymbol{a}|s_h)\eta\mathbb{1}(a_i \neq \pi_{i,h}^\dagger(s))\right]$$

$$= \mathbb{E}_{\pi,\mathbb{A},P}\left[\mathbb{1}(a_{i,h} \neq \pi_{i,h}^\dagger(s))\right]\eta. \tag{43}$$

Thus,

$$\widetilde{V}_{i,1}^{\pi_i^\dagger \times \pi_{-i}}(s_1) - \widetilde{V}_{i,1}^{\pi}(s_1) \geq \mathbb{E}_{\pi,\mathbb{A},P}\left[\sum_{h=1}^{H} \mathbb{1}(a_{i,h} \neq \pi_{i,h}^\dagger(s_h))\right] \eta \geq 0. \tag{44}$$

The target policy is an NE, and then also is an CE or CCE.

Now we prove that the target policy $\pi_i^\dagger$ is the unique {NE, CE, CCE}, when every state $s \in \mathcal{S}$ is reachable at every step $h \in [H]$ under the target policy.

If there exists an CCE $\pi'$ under $\eta$-gap attack, we have $\max_{i \in [m]}(\widetilde{V}_{i,1}^{\dagger,\pi'_{-i}}(s) - \widetilde{V}_{i,1}^{\pi'}(s)) = 0$ for any initial state $s$. If $\pi'_{i,h}(\cdot|s) \neq \pi_{i,h}^\dagger(\cdot|s)$ at a reachable state $s$, we have that $\mathbb{E}_{\pi',\mathbb{A},P}\left[\sum_{h=1}^{H} \mathbb{1}(a_{i,h} \neq \pi_{i,h}^\dagger(s_h))\right]$. Thus,

$$\widetilde{V}_{i,1}^{\pi_i^\dagger \times \pi'_{-i}}(s_1) - \widetilde{V}_{i,1}^{\pi'}(s_1) \geq \mathbb{E}_{\pi',\mathbb{A},P}\left[\sum_{h=1}^{H} \mathbb{1}(a_{i,h} \neq \pi_{i,h}^\dagger(s_h))\right] \eta > 0, \tag{45}$$

and $\pi'$ is not an CCE. In summary, the target policy $\pi_i^\dagger$ is the unique {NE, CE, CCE}.

## H.2 Proof of Theorem 6

From the definition of the best-in-hindsight regret and (51), we have

$$\text{Reg}_i(K, H) = \max_{\pi'_i} \sum_{k=1}^{K}[\widetilde{V}_{i,1}^{\pi'_i \times \pi_{-i}^k}(s_1^k) - \widetilde{V}_{i,1}^{\pi^k}(s_1^k)]$$

$$\geq \sum_{k=1}^{K}[\widetilde{V}_{i,1}^{\pi_i^\dagger \times \pi_{-i}^k}(s_1^k) - \widetilde{V}_{i,1}^{\pi^k}(s_1^k)]. \tag{46}$$

From (44), we have

$$\text{Reg}_i(K, H) \geq \sum_{k=1}^{K} \mathbb{E}_{\pi^k,\mathbb{A},P}\left[\sum_{h=1}^{H} \mathbb{1}[a_{i,h}^k \neq \pi_{i,h}^\dagger(s_h^k)]\right] \eta \tag{47}$$

and

$$\sum_{i=1}^{m} \text{Reg}_i(K, H) = \eta \mathbb{E}[\text{Loss1}(K, H)]. \tag{48}$$

If the best-in-hindsight regret $\text{Reg}(K, H)$ of each agent's algorithm is bounded by a sub-linear bound $\mathcal{R}(T)$, then the attack loss is bounded by $\mathbb{E}[\text{Loss1}(K, H)] \leq m\mathcal{R}(T)/\eta$.

The $\eta$-gap attack strategy attacks all agents when any agent $i$ chooses an non-target action. Note that the rewards are bounded in $[0, 1]$. We have

$$\text{Cost}(K, H) = \sum_{k=1}^{K}\sum_{h=1}^{H}\sum_{i=1}^{m} \left(\mathbb{1}(\widetilde{a}_{i,h}^k \neq a_{i,h}^k) + |\widetilde{r}_{i,h}^k - r_{i,h}^k|\right)$$

$$\leq \sum_{k=1}^{K}\sum_{h=1}^{H}\sum_{i=1}^{m} \mathbb{1}[a_{i,h}^k \neq \pi_{i,h}^\dagger(s_h^k)]m. \tag{49}$$

Hence, the attack cost is bounded by $m\mathbb{E}[\text{Loss1}(K, H)] \leq m^2\mathcal{R}(T)/\eta$.

# I  Analysis of the gray-box attacks

## I.1  Proof of Theorem 7

We assume that the agent does not know the attacker's manipulations and the presence of the attacker. The attacker's manipulations on actions are stationary. We can consider the combination of the attacker and the environment $\text{MG}(\mathcal{S}, \{\mathcal{A}_i\}_{i=1}^m, H, P, \{R_i\}_{i=1}^m)$ as a new environment

$\widetilde{\mathrm{MG}}(\mathcal{S}, \{\mathcal{A}_i\}_{i=1}^m, H, \widetilde{P}, \{\widetilde{R}_i\}_{i=1}^m)$, and the agents interact with the new environment. We define $\widetilde{Q}_i$ and $\widetilde{V}_i$ as the $Q$-values and value functions of the new environment $\widetilde{\mathrm{MG}}$ that each agent $i$ observes.

We first prove that the best response of each agent $i$ towards any policy $\pi_{-i}$ is $\pi_i^\dagger$.

From the mixed attack strategy, we have

$$\widetilde{Q}_{i,h}^\pi(s, \boldsymbol{a}) = \mathbb{1}[a_i = \pi_{i,h}^\dagger(s)] R_{i,h}(s, \pi_h^\dagger(s)) + \mathbb{P}_h \widetilde{V}_{i,h+1}^\pi(s, \pi_h^\dagger(s)). \tag{50}$$

Consider an arbitrary policy $\pi$ and an arbitrary initial state $s_1$. We have

$$\widetilde{V}_{i,1}^{\pi_i^\dagger \times \pi_{-i}}(s_1) - \widetilde{V}_{i,1}^\pi(s_1)$$

$$= \widetilde{V}_{i,1}^{\pi_i^\dagger \times \pi_{-i}}(s_1) - \mathbb{D}_\pi[\widetilde{Q}_{i,1}^{\pi_i^\dagger \times \pi_{-i}}](s_1) + \mathbb{D}_\pi[\widetilde{Q}_{i,1}^{\pi_i^\dagger \times \pi_{-i}}](s_1) - \widetilde{V}_{i,1}^\pi(s_1)$$

$$= \mathbb{E}_{a_{i,1} \sim \pi_{i,1}(\cdot|s_1)} \left[ \mathbb{1}[a_{i,1} \neq \pi_{i,1}^\dagger(s_1)] R_{i,1}(s_1, \pi_1^\dagger(s_1)) \right] + \mathbb{P}_1 \widetilde{V}_{i,2}^{\pi_i^\dagger \times \pi_{-i}}(s_1, \pi_1^\dagger(s_1)) - \mathbb{P}_1 \widetilde{V}_{i,2}^\pi(s_1, \pi_1^\dagger(s_1))$$

$$= \mathbb{E}_{a_{i,1} \sim \pi_{i,1}(\cdot|s_1)} \left[ \mathbb{1}[a_{i,1} \neq \pi_{i,1}^\dagger(s_1)] R_{i,1}(s_1, \pi_1^\dagger(s_1)) \right] + \mathbb{P}_1[\widetilde{V}_{i,2}^{\pi_i^\dagger \times \pi_{-i}} - \widetilde{V}_{i,2}^\pi](s_1, \pi_1^\dagger(s_1))$$

$$= \cdots = \mathbb{E}_{\pi, \mathbb{A}, P} \left[ \sum_{h=1}^H \left( 1 - \pi_{i,h}\left(\pi_{i,h}^\dagger(s_h)|s_h\right) \right) R_{i,h}(s_h, \pi_h^\dagger(s_h)) \right] \geq 0. \tag{51}$$

Since $R_{i,h}(s_h, \pi_h^\dagger(s_h)) > 0$, $\widetilde{V}_{i,1}^{\pi_i^\dagger \times \pi_{-i}}(s_1) - \widetilde{V}_{i,1}^\pi(s_1) = 0$ holds if and only if $\pi_i^\dagger = \pi_i$ holds for the states that are reachable under policy $\pi^\dagger$. We conclude that the best response of each agent $i$ towards any policy $\pi_{-i}$ is $\pi_i^\dagger$ under the mixed attack strategy. The target policy $\pi^\dagger$ is an NE, CE, CCE under the mixed attack strategy.

Now we prove that the target policy $\pi_i^\dagger$ is the unique {NE, CE, CCE} under the mixed attack strategy, when every state $s \in \mathcal{S}$ is reachable at every step $h \in [H]$ under the target policy.

If there exists an CCE $\pi'$ under the mixed attack strategy, we have $\max_{i \in [m]}(\widetilde{V}_{i,1}^{\dagger, \pi'_{-i}}(s) - \widetilde{V}_{i,1}^{\pi'}(s)) = 0$ for any initial state $s$.

From (51), we have that if $\pi'_{i,h}(\cdot|s) \neq \pi_{i,h}^\dagger(\cdot|s)$ at a reachable state $s$, $\widetilde{V}_{i,1}^{\pi_i^\dagger \times \pi'_{-i}}(s_1) - \widetilde{V}_{i,1}^{\pi'}(s_1) > 0$. Then $\widetilde{V}_{i,1}^{\dagger, \pi'_{-i}}(s_1) > \widetilde{V}_{i,1}^{\pi_i^\dagger \times \pi'_{-i}}(s_1) > \widetilde{V}_{i,1}^{\pi'}(s_1)$. $\pi'$ is not an CCE in this case.

We can conclude that $\pi' = \pi^\dagger$ for the states that are reachable under policy $\pi'$. If every state $s \in \mathcal{S}$ is reachable at every step $h \in [H]$ under the target policy, $\pi^\dagger$ is the unique {NE, CE, CCE}.

### I.2 Proof of Theorem 8

We set $R_{min} = \min_{h \in [H]} \min_{s \in \mathcal{S}} \min_{i \in [m]} R_{i,h}(s, \pi_h^\dagger(s))$. From the definition of the best-in-hindsight regret and (51), we have

$$\begin{aligned}
\mathrm{Reg}_i(K, H) &= \max_{\pi'_i} \sum_{k=1}^K [\widetilde{V}_{i,1}^{\pi'_i \times \pi_{-i}^k}(s_1^k) - \widetilde{V}_{i,1}^{\pi^k}(s_1^k)] \\
&\geq \sum_{k=1}^K [\widetilde{V}_{i,1}^{\pi_i^\dagger \times \pi_{-i}^k}(s_1^k) - \widetilde{V}_{i,1}^{\pi^k}(s_1^k)] \\
&= \sum_{k=1}^K \mathbb{E}_{\pi^k, \mathbb{A}, P} \left[ \sum_{h=1}^H \left( 1 - \pi_{i,h}^k\left(\pi_{i,h}^\dagger(s_h^k)|s_h^k\right) \right) R_{i,h}(s_h^k, \pi_h^\dagger(s_h^k)) \right] \\
&= \sum_{k=1}^K \mathbb{E}_{\pi^k, \mathbb{A}, P} \left[ \sum_{h=1}^H \mathbb{1}[a_{i,h}^k \neq \pi_{i,h}^\dagger(s_h^k)] R_{i,h}(s_h^k, \pi_h^\dagger(s_h^k)) \right] \\
&\geq R_{min} \sum_{k=1}^K \mathbb{E}_{\pi^k, \mathbb{A}, P} \left[ \sum_{h=1}^H \mathbb{1}[a_{i,h}^k \neq \pi_{i,h}^\dagger(s_h^k)] \right]
\end{aligned} \tag{52}$$

and

$$\sum_{i=1}^{m} \text{Reg}_i(K, H) \geq R_{min} \mathbb{E}[\text{Loss1}(K, H)]. \tag{53}$$

If the best-in-hindsight regret $\text{Reg}(K, H)$ of each agent's algorithm is bounded by a sub-linear bound $\mathcal{R}(T)$ under the mixed attack strategy, then the attack loss is bounded by $\mathbb{E}[\text{Loss1}(K, H)] \leq m\mathcal{R}(T)/R_{min}$.

The mixed attack strategy only attacks agent $i$ when agent $i$ chooses a non-target action. We have

$$\begin{aligned}
\text{Cost}(K, H) &= \sum_{k=1}^{K} \sum_{h=1}^{H} \sum_{i=1}^{m} \left( \mathbb{1}(\widetilde{a}_{i,h}^k \neq a_{i,h}^k) + |\widetilde{r}_{i,h}^k - r_{i,h}^k| \right) \\
&\leq \sum_{k=1}^{K} \sum_{h=1}^{H} \sum_{i=1}^{m} \mathbb{1}[\widetilde{a}_{i,h}^k \neq a_{i,h}^k] \, (1 + 1) \, .
\end{aligned} \tag{54}$$

Then, the attack cost is bounded by $2\mathbb{E}[\text{Loss1}(K, H)] \leq 2m\mathcal{R}(T)/R_{min}$.

## J  Analysis of the black-box attacks

### J.1  Proof of Lemma 1

We denote by $\overline{Q}_{\dagger,h}^k$, $\underline{Q}_{\dagger,h}^k$, $\overline{V}_{\dagger,h}^k$ $\underline{V}_{\dagger,h}^k$, $N_h^k$, $\hat{\mathbb{P}}_h^k$, $\pi_h^k$ and $\hat{R}_{\dagger,h}^k$ the observations of the approximate mixed attacker at the beginning of episode $k$ and time step $h$. As before, we begin with proving that the estimations are indeed upper and lower bounds of the corresponding $Q$-values and state value functions. We use $\pi^*$ to denote the optimal policy that maximizes the attacker's rewards, i.e. $V_{\dagger,1}^{\pi^*}(s) = \max_\pi V_{\dagger,1}^\pi(s)$.

**Lemma 2** *With probability $1 - p$, for any $(s, \boldsymbol{a}, h)$ and $k \leq \tau$,*

$$\overline{Q}_{\dagger,h}^k(s, \boldsymbol{a}) \geq Q_{\dagger,h}^{\pi^*}(s, \boldsymbol{a}), \ \underline{Q}_{\dagger,h}^k(s, \boldsymbol{a}) \leq Q_{\dagger,h}^{\pi^k}(s, \boldsymbol{a}), \tag{55}$$

$$\overline{V}_{\dagger,h}^k(s) \geq V_{\dagger,h}^{\pi^*}(s), \ \underline{V}_{\dagger,h}^k(s) \leq V_{\dagger,h}^{\pi^k}(s). \tag{56}$$

**Proof.** For each fixed $k$, we prove this by induction from $h = H + 1$ to $h = 1$. For the step $H + 1$, we have $\overline{V}_{\dagger,H+1}^k = \underline{V}_{\dagger,H+1}^k = Q_{\dagger,H+1}^{\pi^*} = \boldsymbol{0}$. Now, we assume inequality (56) holds for the step $h + 1$. By the definition of $Q$-values and Algorithm 1, we have

$$\begin{aligned}
&\overline{Q}_{\dagger,h}^k(s, \boldsymbol{a}) - Q_{\dagger,h}^{\pi^*}(s, \boldsymbol{a}) \\
&= \hat{R}_{\dagger,h}^k(s, \boldsymbol{a}) - R_{\dagger,h}^k(s, \boldsymbol{a}) + \hat{\mathbb{P}}_h^k \overline{V}_{\dagger,h+1}^k(s, \boldsymbol{a}) - \mathbb{P}_h V_{i,h}^{\pi^*}(s, \boldsymbol{a}) + B(N_h^k(s, \boldsymbol{a})) \\
&= \hat{\mathbb{P}}_h^k(\overline{V}_{\dagger,h+1}^k - V_{\dagger,h}^{\pi^*})(s, \boldsymbol{a}) + (\hat{R}_{\dagger,h}^k - R_{\dagger,h}^k)(s, \boldsymbol{a}) + (\hat{\mathbb{P}}_h^k - \mathbb{P}_h)V_{\dagger,h}^{\pi^*}(s, \boldsymbol{a}) + B(N_h^k(s, \boldsymbol{a})).
\end{aligned} \tag{57}$$

Recall that $B(N) = (H\sqrt{S} + 1)\sqrt{\log(2AH\tau/p)/(2N)}$. By Azuma-Hoeffding inequality, we have that with probability $1 - 2p/SAH$,

$$\forall k \leq \tau, \left| \hat{R}_{\dagger,h}^k(s, \boldsymbol{a}) - R_{\dagger,h}^k(s, \boldsymbol{a}) \right| \leq \sqrt{\frac{\log(2SAH\tau/p)}{2N_h^k(s, \boldsymbol{a})}}, \tag{58}$$

and

$$\forall k \leq \tau, \left| (\hat{\mathbb{P}}_h^k - \mathbb{P}_h)V_{\dagger,h}^{\pi^*}(s, \boldsymbol{a}) \right| \leq H\sqrt{\frac{S\log(2SAH\tau/p)}{2N_h^k(s, \boldsymbol{a})}}. \tag{59}$$

Putting everything together, we have $\overline{Q}_{\dagger,h}^k(s, \boldsymbol{a}) - Q_{\dagger,h}^{\pi^*}(s, \boldsymbol{a}) \geq \hat{\mathbb{P}}_h^k(\overline{V}_{\dagger,h+1}^k - V_{\dagger,h}^{\pi^*})(s, \boldsymbol{a}) \geq 0$. Similarly, $\underline{Q}_{\dagger,h}^k(s, \boldsymbol{a}) \leq Q_{\dagger,h}^{\pi^k}(s, \boldsymbol{a})$.

Now we assume inequality (55) holds for the step $h$. As discussed above, if inequality (56) holds for the step $h+1$, inequality (55) holds for the step $h$. By Algorithm 1, we have

$$\overline{V}_{\dagger,h}^k(s) = \overline{Q}_{\dagger,h}^k(s, \pi_h^k(s)) \geq \overline{Q}_{\dagger,h}^k(s, \pi_h^*(s)) \geq Q_{\dagger,h}^{\pi^*}(s, \pi_h^*(s)) = V_{\dagger,h}^{\pi^*}(s). \tag{60}$$

Similarly, $\underline{V}_{\dagger,h}^k(s) \leq V_{\dagger,h}^{\pi^k}(s)$. $\qquad\square$

Now, we are ready to prove Lemma 1. By Azuma-Hoeffding inequality, we have that with probability $1 - 2p$,

$$\left| \left( \mathbb{E}_{s_1 \sim P_0(\cdot)} - \mathbb{E}_{s_1 \sim \hat{\mathbb{P}}_0(\cdot)} \right) \left[ V_{\dagger,1}^{\pi^*}(s_1) - V_{\dagger,1}^{\pi^k}(s_1) \right] \right| \leq H\sqrt{\frac{S \log(2\tau/p)}{2k}}, \tag{61}$$

and $\forall k \leq \tau$,

$$\left| \sum_{k'=1}^k \left( \mathbb{E}_{s_1 \sim P_0(\cdot)} - \mathbb{1}(s_1 = s_1^{k'}) \right) \left[ V_{\dagger,1}^{\pi^*}(s_1) - V_{\dagger,1}^{\pi^{k'}}(s_1) \right] \right| \leq H\sqrt{\frac{S \log(2\tau/p)}{2k}}. \tag{62}$$

Thus, for any $k \leq \tau$,

$$\mathbb{E}_{s_1 \sim P_0(\cdot)} \left[ V_{\dagger,1}^{\pi^*}(s_1) - V_{\dagger,1}^{\pi^k}(s_1) \right]$$
$$\leq \mathbb{E}_{s_1 \sim \hat{\mathbb{P}}_0^k(\cdot)} \left[ V_{\dagger,1}^{\pi^*}(s_1) - V_{\dagger,1}^{\pi^k}(s_1) \right] + H\sqrt{S \log(2\tau/p)/(2k)} \tag{63}$$
$$\leq \mathbb{E}_{s_1 \sim \hat{\mathbb{P}}_0^k(\cdot)} \left[ \overline{V}_{\dagger,1}^k(s_1) - \underline{V}_{\dagger,1}^k(s_1) \right] + H\sqrt{S \log(2\tau/p)/(2k)}.$$

According to (61) and (62), we have

$$\sum_{k=1}^\tau \left( \mathbb{E}_{s_1 \sim \hat{\mathbb{P}}_0^k(\cdot)} \left[ \overline{V}_{\dagger,1}^k(s_1) - \underline{V}_{\dagger,1}^k(s_1) \right] + H\sqrt{S \log(2\tau/p)/(2k)} \right)$$
$$\leq \sum_{k=1}^\tau \left( \overline{V}_{\dagger,1}^k(s_1^k) - \underline{V}_{\dagger,1}^k(s_1^k) \right) + \sum_{k=1}^\tau 3H\sqrt{S \log(2\tau/p)/(2k)}. \tag{64}$$

We define $\Delta V_h^k(s) = \overline{V}_{\dagger,h}^k(s) - \underline{V}_{\dagger,h}^k(s)$, $\Delta Q_h^k(s, \boldsymbol{a}) = \overline{Q}_{\dagger,h}^k(s, \boldsymbol{a}) - \underline{Q}_{\dagger,h}^k(s, \boldsymbol{a})$. By the update equations in Algorithm 1, we have $\Delta Q_h^k(s, \boldsymbol{a}) \leq \hat{\mathbb{P}}_h^k \Delta V_{h+1}^k(s, \boldsymbol{a}) + 2B(N_h^k(s, \boldsymbol{a}))$ and $\Delta V_h^k(s) = \Delta Q_h^k(s, \pi_h^k(s))$. We define $\psi_h^k = \Delta V_h^k(s_h^k) = \Delta Q_h^k(s_h^k, a_h^k)$. From (59) and (66), we have

$$\psi_h^k \leq \hat{\mathbb{P}}_h^k \Delta V_{h+1}^k(s_h^k, \boldsymbol{a}_{\boldsymbol{h}}^k) + 2B(N_h^k(s_h^k, \boldsymbol{a}_h^k))$$
$$\leq \mathbb{P}_h^k \Delta V_{h+1}^k(s_h^k, \boldsymbol{a}_{\boldsymbol{h}}^k) + 3B(N_h^k(s_h^k, \boldsymbol{a}_h^k)) \tag{65}$$
$$\leq \mathbb{P}_h^k \Delta V_{h+1}^k(s_h^k, \boldsymbol{a}_{\boldsymbol{h}}^k) - \psi_{h+1}^k + \psi_{h+1}^k + 3B(N_h^k(s_h^k, \boldsymbol{a}_h^k)).$$

By Azuma-Hoeffding inequality, we have that with probability $1 - p/H$, $\forall k \leq \tau$,

$$\left| \sum_{k'=1}^k |\mathbb{P}_h^{k'} \Delta V_{h+1}^k(s_h^{k'}, \boldsymbol{a}_{\boldsymbol{h}}^{k'}) - \psi_{h+1}^{k'}| \right| \leq H\sqrt{\frac{S \log(2H\tau/p)}{2k}}. \tag{66}$$

Since $\psi_{H+1}^k = 0$ for all $k$, we have

$$\sum_{k=1}^\tau \psi_1^k \leq \sum_{k=1}^\tau \sum_{h=1}^H |\mathbb{P}_h^k \Delta V_{h+1}^k(s_h^k, \boldsymbol{a}_{\boldsymbol{h}}^k) - \psi_{h+1}^k| + \sum_{k=1}^\tau \sum_{h=1}^H 3B(N_h^k(s_h^k, \boldsymbol{a}_h^k))$$
$$\leq \sum_{h=1}^H H\sqrt{\frac{S \log(2H\tau/p)}{2\tau}} + \sum_{h=1}^H \sum_{(s, \boldsymbol{a})} \sum_{n=1}^{N_h^\tau(s, \boldsymbol{a})} (H\sqrt{S} + 1)\sqrt{\frac{\log(2SAH\tau/p)}{2n}} \tag{67}$$
$$\leq H^2 \sqrt{\frac{S \log(2H\tau/p)}{2\tau}} + H(H\sqrt{S} + 1)\sqrt{2SA\tau \log(2SAH\tau/p)}$$

and therefore

$$\sum_{k=1}^{\tau} \left( \mathbb{E}_{s_1 \sim \hat{\mathbb{P}}_0^k(\cdot)} \left[ \overline{V}_{\dagger,1}^k(s_1) - \underline{V}_{\dagger,1}^k(s_1) \right] + H\sqrt{S\log(2\tau/p)/(2k)} \right)$$

$$\leq H^2 \sqrt{\frac{S\log(2H\tau/p)}{2\tau}} + 3H\sqrt{2S\tau\log(2\tau/p)} + H(H\sqrt{S}+1)\sqrt{2SA\tau\log(2SAH\tau/p)}. \tag{68}$$

Since

$$\pi^{\dagger} = \min_{\pi_k} \left( \mathbb{E}_{s_1 \sim \hat{\mathbb{P}}_0^k(\cdot)} \left[ \sum_{i=1}^{m} \left( V_{\dagger,1}^{\pi^*}(s_1) - V_{\dagger,1}^{\pi^k}(s_1) \right) \right] + H\sqrt{S\log(2\tau/p)/(2k)} \right), \tag{69}$$

$$\mathbb{E}_{s_1 \sim P_0(\cdot)} \left[ V_{\dagger,1}^{\pi^*}(s_1) - V_{\dagger,1}^{\pi^{\dagger}}(s_1) \right]$$

$$\leq \mathbb{E}_{s_1 \sim \hat{\mathbb{P}}_0^k(\cdot)} \left[ V_{\dagger,1}^{\pi^*}(s_1) - V_{\dagger,1}^{\pi^{\dagger}}(s_1) \right] + H\sqrt{S\log(2\tau/p)/(2k)}$$

$$\leq H^2 \sqrt{\frac{S\log(2H\tau/p)}{2\tau^3}} + 3H\sqrt{2S\log(2\tau/p)/\tau} + H(H\sqrt{S}+1)\sqrt{2SA\log(2SAH\tau/p)/\tau} \tag{70}$$

$$\leq 2H^2 S\sqrt{2A\log(2SAH\tau/p)/\tau},$$

where the last inequality holds when $S, H, A \geq 2$. Similarly,

$$\sum_{k=1}^{K} \left[ V_{\dagger,1}^{\pi^*}(s_1^k) - V_{\dagger,1}^{\pi^{\dagger}}(s_1^k) \right] \leq 2H^2 S\sqrt{2A\log(2SAH\tau/p)/\tau}. \tag{71}$$

## J.2 Proof of Theorem 9

For completeness, we describe the main steps of V-learning algorithm [Jin et al., 2021] in Algorithm 2 and the adversarial bandit algorithm in Algorithm 3.

---

**Algorithm 2:** V-learning [Jin et al., 2021]

---

1: For any $(s, a, h)$, $V_h(s) \leftarrow H + 1 - h$, $N_h(s) \leftarrow 0$, $\pi_h(a|s) \leftarrow 1/A$.
2: **for** episodes $k = 1, \ldots, K$ **do**
3:    receive $s_1$
4:    **for** episodes $h = 1, \ldots, H$ **do**
5:       take action $a_h \sim \pi_h(\cdot|s_h)$, observe reward $r_h$ and next state $s_{h+1}$.
6:       $t = N_h(s_h) \leftarrow N_h(s_h) + 1$.
7:       $\overline{V}_h(s_h) \leftarrow (1 - \alpha_t)\overline{V}_h(s_h) + \alpha_t(rh + V_{h+1}(s_{h+1}) + \beta_t)$.
8:       $V_h(s_h) \leftarrow \min\{H + 1 - h, \overline{V}_h(s_h)\}$
9:       $\pi_h(\cdot|s_h) \leftarrow$ ADV_BANDIT_UPDATE$(a_h, \frac{H - r_h - V_{h+1}(s_{h+1})}{H})$ on $(s_h, h)^{th}$ adversarial bandit.
10:   **end for**
11: **end for**

---

---

**Algorithm 3:** FTRL for Weighted External Regret [Jin et al., 2021]

---

1: For any $b \in \mathcal{B}$, $\theta_1(b) \leftarrow 1/B$.
2: **for** episode $t = 1, \ldots, K$ **do**
3:    Take action $b_t \sim \theta_t(\cdot)$, and observe loss $\widetilde{l}_t(b_t)$.
4:    $\hat{l}_t(b) \leftarrow \widetilde{l}_t(b_t)\mathbb{1}[b_t = b]/(\theta_t(b) + \gamma_t)$ for all $b \in \mathcal{B}$.
5:    $\theta_{t+1}(b) \propto \exp[-(\gamma_t/w_t)\sum_{i=1}^{t} w_i \hat{l}_i(b)]$
6: **end for**

---

We use the same learning rate $\alpha_t$ in [Jin et al., 2021]. We also use an auxiliary sequence $\{\alpha_t^i\}_{i=1}^t$ defined in [Jin et al., 2021] based on the learning rate, which will be frequently used in the proof:

$$\alpha_t = \frac{H+1}{H+t}, \ \alpha_t^0 = \prod_{j=1}^{t}(1 - \alpha_j), \ \alpha_t^i = \alpha_i \prod_{j=i+1}^{t}(1 - \alpha_j). \tag{72}$$

We follow the requirement for the adversarial bandit algorithm used in V-learning, which is to have a high probability weighted external regret guarantee as follows.

**Assumption 1** *For any $t \in \mathbb{N}$ and any $\delta \in (0,1)$, with probability at least $1 - \delta$, we have*

$$\max_{\theta} \sum_{j=1}^{t} \alpha_t^j [\langle \theta_j, l_j \rangle - \langle \theta, l_j \rangle] \leq \xi(B, t, \log(1/\delta)). \tag{73}$$

*In addition, there exists an upper bound $\Xi(B, t, \log(1/\delta)) \geq \sum_{t'=1}^{t} \xi(B, t, \log(1/\delta))$ where (i) $\xi(B, t, \log(1/\delta))$ is non-decreasing in B for any t, $\delta$; (ii) $\Xi(B, t, \log(1/\delta))$ is concave in t for any B, $\delta$.*

In particular, it was proved in [Jin et al., 2021] that the Follow-the-Regularized-Leader (FTRL) algorithm (Algorithm 3) satisfies Assumption 1 with bounds $\xi(B, t, \log(1/\delta)) \leq \mathcal{O}(\sqrt{HB \log(B/\delta)/t})$ and $\Xi(B, t, \log(1/\delta)) \leq \mathcal{O}(\sqrt{HBt \log(B/\delta)})$. By choosing hyper-parameter $w_t = \alpha_t \left( \prod_{i=2}^{t} (1 - \alpha_i) \right)^{-1}$ and $\gamma_t = \sqrt{\frac{H \log B}{Bt}}$, $\xi(B, t, \log(1/\delta)) = 10\sqrt{HB \log(B/\delta)/t}$ and $\Xi(B, t, \log(1/\delta)) = 20\sqrt{HBt \log(B/\delta)}$.

We use $V^k$, $N^k$, $\pi^k$ to denote the value, counter and policy maintained by V-learning algorithm at the beginning of the episode $k$. Suppose $s$ was previously visited at episodes $k^1, \cdots, k^t < k$ at the step $h$. Set $t'$ such that $k^{t'} \leq \tau$ and $k^{t'+1} > \tau$.

In the exploration phase of the proposed approximate mixed attack strategy, the rewards are equal to 1 for any state $s$, any action $\boldsymbol{a}$, any agent $i$ and any step $h$. The loss updated to the adversarial bandit update step in Algorithm 2 is equal to $\frac{h-1}{H}$.

In the attack phase, the expected loss updated to the adversarial bandit update step in Algorithm 2 is equal to

$$\sum_{j=1}^{t'} \alpha_t^j \frac{h-1}{H} + \sum_{j=t'+1}^{t} \alpha_t^j \mathbb{D}_{\pi^\dagger} \left( \frac{H - \mathbb{P}_h V_{i,h+1}^{k^j}}{H} \right)(s) + \sum_{j=t'+1}^{t} \alpha_t^j \mathbb{D}_{\pi_h^{k^j}} \left( \frac{-\widetilde{r}_{i,h}}{H} \right)(s). \tag{74}$$

Thus, in both of the exploration phase and the attack phase, $\pi^\dagger$ is the best policy for the adversarial bandit algorithm.

By Assumption 1 and the adversarial bandit update step in Algorithm 2, with probability at least $1 - \delta$, for any $(s, h) \in \mathcal{S} \times [H]$ and any $k > \tau$, we have

$$\begin{aligned}
\xi(A, t, \iota) \geq & \sum_{j=1}^{t'} \alpha_t^j \frac{h-1}{H} + \sum_{j=t'+1}^{t} \alpha_t^j \mathbb{D}_{\pi^\dagger} \left( \frac{H - \mathbb{P}_h V_{i,h+1}^{k^j}}{H} \right)(s) + \sum_{j=t'+1}^{t} \alpha_t^j \mathbb{D}_{\pi_h^{k^j}} \left( \frac{-\widetilde{r}_{i,h}}{H} \right)(s) \\
& - \sum_{j=1}^{t'} \alpha_t^j \frac{h-1}{H} + \sum_{j=t'+1}^{t} \alpha_t^j \mathbb{D}_{\pi^\dagger} \left( \frac{H - \mathbb{P}_h V_{i,h+1}^{k^j}}{H} \right)(s) + \sum_{j=t'+1}^{t} \alpha_t^j \mathbb{D}_{\pi^\dagger} \left( \frac{-\widetilde{r}_{i,h}}{H} \right)(s) \\
= & \sum_{j=t'+1}^{t} \alpha_t^j \left( 1 - \pi_{i,h}^{k^j}(\pi_{i,h}^\dagger(s)|s) \right) \frac{r_{i,h}(s, \pi_h^\dagger(s))}{H},
\end{aligned} \tag{75}$$

where $\iota = \log(mHSAK/\delta)$.

Note that $R_{min} = \min_{h \in [H]} \min_{s \in \mathcal{S}} \min_{i \in [m]} R_{i,h}(s, \pi_h^\dagger(s))$. We have

$$\frac{H}{R_{min}} \xi(A, t, \iota) \geq \sum_{j=t'+1}^{t} \alpha_t^j \left( 1 - \pi_{i,h}^{k^j}(\pi_{i,h}^\dagger(s)|s) \right). \tag{76}$$

Let $n_h^k = N_h^k(s_h^k)$ and suppose $s_h^k$ was previously visited at episodes $k^1, \cdots, k^{n_h^k} < k$ at the step $h$. Let $k^j(s)$ denote the episode that $s$ was visited in $j$-th time.

$$\frac{H}{R_{min}} \xi(A, n_h^k, \iota) \geq \sum_{j=N_h^\tau(s_h^k)+1}^{n_h^k} \alpha_{n_h^k}^j \left( 1 - \pi_{i,h}^{k^j}(\pi_{i,h}^\dagger(s_h^k)|s_h^k) \right). \tag{77}$$

According to the property of the learning rate $\alpha_t$, we have

$$\frac{H}{R_{min}\alpha_t^t}\xi(A, n_h^k, \iota) + \sum_{j=N_h^\tau(s_h^k)+1}^{n_h^k-1} \frac{H}{R_{min}}\xi(A, j, \iota) \leq \sum_{j=N_h^\tau(s_h^k)+1}^{n_h^k} \left(1 - \pi_{i,h}^{k^j}(\pi_{i,h}^\dagger(s_h^k)|s_h^k)\right). \quad (78)$$

Then,

$$\frac{40H}{R_{min}}\sqrt{HAn_h^k\iota} \leq \sum_{j=N_h^\tau(s_h^k)+1}^{n_h^k} \left(1 - \pi_{i,h}^{k^j}(\pi_{i,h}^\dagger(s_h^k)|s_h^k)\right). \quad (79)$$

Computing the summation of the above inequality over $h$ and $s$, we have

$$\begin{aligned}
\mathbb{E}&\left[\sum_{h=1}^H \sum_{k=\tau+1}^K \mathbb{1}[a_{i,h}^k \neq \pi_{i,h}^\dagger(s_h^k)]\right] \\
&= \sum_{h=1}^H \sum_{s\in\mathcal{S}} \sum_{j=N_h^\tau(s)+1}^{N_h^K(s)} \left(1 - \pi_{i,h}^{k^j(s)}(\pi_{i,h}^\dagger(s)|s)\right) \\
&\leq \sum_{s\in\mathcal{S}} \frac{40}{R_{min}}\sqrt{H^5AN_h^K(s)\iota} \\
&\leq \frac{40}{R_{min}}\sqrt{H^5ASK\iota}.
\end{aligned} \quad (80)$$

In the exploration phase, the loss at each episode is up to $H$. In the attack phase, the expected number of episodes that the agents do not follow $\pi^\dagger$ is up to $\frac{40}{R_{min}}m\sqrt{H^7ASK\iota}$.

According to Lemma 1, the attack loss is bounded by

$$\mathbb{E}\left[\text{loss}(K, H)\right] \leq H\tau + \frac{40}{R_{min}}m\sqrt{H^9ASK\iota} + 2H^2SK\sqrt{2A\iota/\tau}. \quad (81)$$

In the exploration phase, the approximate mixed attack strategy attacks at any step and any episode. In the attack phase, the approximate mixed attack strategy only attacks agent $i$ when agent $i$ chooses a non-target action. We have

$$\begin{aligned}
\text{Cost}(K, H) &= \sum_{k=1}^K \sum_{h=1}^H \sum_{i=1}^m \left(\mathbb{1}(\widetilde{a}_{i,h}^k \neq a_{i,h}^k) + |\widetilde{r}_{i,h}^k - r_{i,h}^k|\right) \\
&\leq \sum_{k=1}^\tau \sum_{h=1}^H \sum_{i=1}^m (1+1) + \sum_{k=\tau=1}^K \sum_{h=1}^H \sum_{i=1}^m \mathbb{1}[\widetilde{a}_{i,h}^k \neq a_{i,h}^k](1+1).
\end{aligned} \quad (82)$$

Then, the attack cost is bounded by

$$\mathbb{E}\left[\text{Cost}(K, H)\right] \leq 2mH\tau + \frac{80}{R_{min}}\sqrt{H^5ASK\iota}. \quad (83)$$

For the executing output policy $\hat{\pi}$ of V-learning, we have

$$1 - \hat{\pi}_{i,h}(\pi_{i,h}^\dagger(s)|s)$$

$$= \frac{1}{K} \sum_{k=1}^{K} \sum_{j=1}^{N_h^k(s)} \alpha_{N_h^k(s)}^j \left(1 - \pi_{i,h}^{k^j(s)}(\pi_{i,h}^\dagger(s)|s)\right)$$

$$= \frac{1}{K} \sum_{k=\tau+1}^{K} \sum_{j=N_h^\tau(s)+1}^{N_h^k(s)} \alpha_{N_h^k(s)}^j \left(1 - \pi_{i,h}^{k^j(s)}(\pi_{i,h}^\dagger(s)|s)\right) + \frac{1}{K} \sum_{k=1}^{\tau} \sum_{j=1}^{N_h^k(s)} \alpha_{N_h^k(s)}^j \left(1 - \pi_{i,h}^{k^j(s)}(\pi_{i,h}^\dagger(s)|s)\right)$$

$$+ \frac{1}{K} \sum_{k=\tau+1}^{K} \sum_{j=1}^{N_h^\tau(s)} \alpha_{N_h^k(s)}^j \left(1 - \pi_{i,h}^{k^j(s)}(\pi_{i,h}^\dagger(s)|s)\right)$$

$$\leq \frac{20}{R_{min}} \sqrt{\frac{H^3 A \iota}{K}} + \frac{2\tau}{K}.$$

(84)

The probability that the agents with $\hat{\pi}$ do not follow the target policy is bounded by $\frac{20mS}{R_{min}} \sqrt{\frac{H^5 A \iota}{K}} + \frac{2\tau mSH}{K}$.

According to Lemma 1, the attack loss of the executing output policy $\hat{\pi}$ is upper bounded by

$$V_{\dagger,1}^{\pi^*}(s_1) - V_{\dagger,1}^{\hat{\pi}}(s_1) \leq H \left( \frac{20mS}{R_{min}} \sqrt{\frac{H^5 A \iota}{K}} + \frac{2\tau mSH}{K} \right) + 2H^2 S \sqrt{2A\iota/\tau}. \tag{85}$$

