# OpenReview forum: "Efficient Adversarial Attacks on Online Multi-agent Reinforcement Learning"
_NeurIPS.cc/2023/Conference — NeurIPS 2023 poster_

### Official Review · Reviewer_Mdqm · 2023-06-26

**Soundness:** 3 good
**Presentation:** 3 good
**Contribution:** 3 good
**Rating:** 6
**Confidence:** 3

**Summary:**

This paper proposes a series of novel attacks against MARL under different assumptions. From the theoretical perspective, it then analyzes the efficacy of the proposed attacks.

**Strengths:**

+ The paper proposes a series of novel attacks against online MARL and provides a decent theoretical analysis for the efficacy of the proposed attack. This paper lays a theoretical foundation for the adversarial attacks against online MARL. The proposed attacks can be potentially extended by follow-up works to launch practical attacks against MARL systems. The novelty of the problem and the technical depth are enough for a top-tier ML conference.

**Weaknesses:**

I do not idenitfy any critical weakness in the proposed technique and the corresponding theoretical analysis. As a theoretical paper, it is a little bit over to ask questions about the practical perspective.  However, as an attack paper, it is somewhat important to discuss the practicability of the proposed attacks. As such, I would suggest the authors add such a discussion related to the proposed technique. For example, the authors could provide some suggestions for practitioners to solve the proposed objective functions, which could be hard to optimize. In addition, what would be the computational cost for the proposed objective functions?

**Questions:**

1. Why the proposed assumptions that the attackers can freely alter the target agents' actions and rewards are practical and realistic?
2. What are the practical suggestions for optimizing the proposed objective functions?
3. What is the computational cost of solving the proposed attack objective functions?

**Limitations:**

This paper does not include a section discussing the limitation of the proposed attacks. Given that the paper is mainly about theoretical analysis of attack efficacy. It could include a discussion section (maybe in the appendix) to discuss the potential limitations maybe on the practical side, i.e., how realistic are the assumptions that the attackers can freely manipulate the action and reward? What is the computational cost of solving the proposed attack objective functions?

In addition, the authors checked the broader impact of the checklist. However, I do not find a (sub)section or paragraph clearly addressing the potential negative social impact.

---

> ### Author Rebuttal · Authors · 2023-08-10
>
> > Why the proposed assumptions that the attackers can freely alter the target agents' actions and rewards are practical and realistic?
>
> In some applications of RL models, action decisions and reward signals may need to be transmitted over communication links. When data packets containing the reward signals and action decisions etc are transmitted through the communication links, an attacker can implement adversarial attacks by intercepting and modifying these data packets. Hence, the proposed adversarial attacks are possible and realistic.
>
> As the first step of understanding the potential risks of different adversarial attacks on online MARL, we did not limit the attacker's abilities. The attackers can freely alter the target agents' actions and rewards. There may exist some situation that the attacker can not freely alter any actions and rewards and can only manipulate part of actions or rewards. Considering limited attacker and finding the defending algorithms is an important future direction for us to pursue.
>
> > What are the practical suggestions for optimizing the proposed objective functions?
>
> In the practical side, the proposed mixed attack strategy works for many realistic problems. The mixed attack strategy does not require any information about the underline Markov game. It is simple yet effective. As the first step of understanding the impact of adversarial attacks on online MARL, our analysis is focused on the tabular case. The proposed mixed attack strategy can work in some realistic problems where the action space is continuous. In the continuous action problems, the attacker does not aim to force the agents to learn the exactly target policy but some policies nearby the target policy. The proposed mixed attack strategy still works. For example, in the continuous space, the attacker can change the non-target action to $r_{i.h}*e^{c |a_{i,h}-a^+_{i,h}|}$ instead of $0$, in order to avoiding sparse reward. Then the agents will still learn a policy that is close to the target policy.
>
> > What is the computational cost of solving the proposed attack objective functions?
>
> For the proposed black-box attack strategy (the approximation mixed attack), the computational cost is $O(S^2AH\tau+mKH)$. The proposed algorithm will compute the $Q$-values for each visited action-state pair at every steps and every episodes in the exploration phase. The computation of $Q$-value costs $O(S)$. Thus, the total computational cost in the exploration phase is $O(S^2AH\tau)$. In the attack phase, the attacker only need to change the action and the reward for each agent so that the computational cost in the attack phase is $O(mKH)$. We will add this discussion to the revised paper.

---

> > ### Comment · Reviewer_Mdqm · 2023-08-15
> >
> > Thank the authors for the clarification. I do not have further questions. Please add the rebuttal changes to the paper.

---

> > > ### Author Response · Authors · 2023-08-17
> > > **Thanks!**
> > >
> > > We thank the reviewer again for the valuable feedback and the time invested. We will definitely add the rebuttal changes to the paper.

---

### Official Review · Reviewer_qHBd · 2023-07-05

**Soundness:** 2 fair
**Presentation:** 2 fair
**Contribution:** 2 fair
**Rating:** 5
**Confidence:** 2

**Summary:**

This paper presents a novel approach to adversarial attacks in Multi-agent Reinforcement Learning (MARL), offering significant insights into various attack settings.  The authors explore different attack strategies, including reward poisoning, action poisoning, and a combination of both, across various settings: white-box, grey-box, and black-box. The paper's key contribution is the efficient adversarial attack strategy that perturbs both the action and reward of the agents.

**Strengths:**

- The paper addresses a novel and significant problem of adversarial attacks in Multi-agent Reinforcement Learning (MARL) for reward/action poisoning.
- The paper is  moderately easy to follow.
- The authors have considered different settings such as white-box, grey-box, and black-box, which adds depth to the study.


**Weaknesses:**

- The related work section of the paper does not seem to cover the literature comprehensively. It would be beneficial for the authors to provide a more thorough review of the existing literature, especially focusing on the works that have addressed similar problems in the past, how they did it (for reward/attack poisoning).
- The paper lacks numerical results. While the authors have provided some simulation results in the appendix, it would be beneficial to include more numerical results in the main body of the paper to strengthen their arguments. Furthermore, this is a quite practical problem, and it is  beneficial to thoroughly  assess the concepts introduced by the authors by means of numerical simulations.
- Attack detection is not discussed by the authors.
-The authors have analyzed the performance of the proposed attack algorithm on V-learning. It would be interesting to see how the proposed black-box strategy performs on other MARL algorithms.
- It's not clear how tight the bounds proposed by the authors are.

**Questions:**

See above

**Limitations:**

Authors do not seem to discuss limitations thoroughly.

---

> ### Author Rebuttal · Authors · 2023-08-10
>
> > The related work section of the paper does not seem to cover the literature comprehensively. It would be beneficial for the authors to provide a more thorough review of the existing literature, especially focusing on the works that have addressed similar problems in the past, how they did it (for reward/attack poisoning).
>
> Thank you very much for this comment. We will add the following discussions to the revised paper:
>
> [Ma et al., 2019] studies reward poisoning attack against batch RL in which the attacker is able to gather and modify the collected batch data. [Rakhsha et al., 2020] proposes a white-box environment poisoning model in which the attacker could manipulate the original MDP to a poisoned MDP. [Behzadan and Munir, 2017, Zhang et al., 2020., Rangi et al., 2022] study online white-box reward poisoning attacks in which the attacker could manipulate the reward signal before the agent receives it. [Sun et al., 2021] proposes a practical black-box poisoning algorithm called VA2C-P. Their empirical results show that VA2C-P works for deep policy gradient RL agents without any prior knowledge of the environment. [Rakhsha et al., 2021] develops a black-box reward poisoning attack strategy called U2, that can provably attack any efficient RL algorithms. [Xu et al., 2021] investigates training-time attacks on RL agents and the introduced attacker can manipulate the environment. [Wang et al., 2021] studies the backdoor attack in two-player competitive RL systems. The trigger is the action of another agent in the environment. They propose a unified method to design fast-failing agents which will fast fail when trigger occurred. [Liu et al., 2022] studies the controllable attack by constraining the state distribution shift caused by the adversarial policy and offering a more controllable attack scheme. [chen et al., 2022] considers a situation that $\alpha$-fraction of agents are adversarial and can report arbitrary fake information. They design two Byzantine-robust distributed  value iteration algorithms that can identify a near-optimal policy with near-optimal sample complexity. [Mohammadi et al., 2023] studies targeted poisoning attacks in a two-agent setting where an attacker implicitly poisons the effective environment of one of the agents by modifying the policy of its peer. [Ma et al., 2022] considers a game redesign problem where the designer knows the full information of the game and can redesign the reward functions. The proposed redesign methods can incentivize players to take a specific target action profile frequently with a small cumulative design cost. [Gleave et al., 2020, Guo et al., 2021] studies the poisoning attack on multi agent reinforcement learners, assuming that the attacker controls one of the learners. [Wu et al., 2022] studies the reward poisoning attack on offline multi-agent reinforcement learners.
>
> > The paper lacks numerical results.
>
> Due to the page limitation, we put the Numerical results in Appendix B. In addition, in the author rebuttal pdf file, we also added more experimental results to discuss how the attack loss and cost are affected by the environment's parameters.
>
> > Attack detection is not discussed by the authors.
>
> We agree with the reviewer's comment that we did not consider the attack detection problem, which is certainly very important.
> In this paper, we assumed that the agents do not know the existence of the attacker. In fact, in the online setting, if the agents have no prior information of the MG, the proposed white and gray box attack is hard to be detected. As we consider the Markov attack strategy in this paper, the post-attack environment under the Markov attack strategy is still a Markov game. Without reference, the agents can not figure out whether the environment they observe is a post-attack environment or an attack-free environment. The proposed black attack may be detected, as the transition probabilities of the post-attack environment change over time.
> The goal of our paper is to understand and identify the impacts of different adversarial attacks. We hope our work can inspire follow-up work that can detect and mitigate such attacks so that RL models can be used in safety-critical applications.
> It is an important future direction for us to pursue. We will add these discussions in the revised paper.
>
> > It's not clear how tight the bounds proposed by the authors are.
>
> Currently, we are not able to give a information-theoretic lower bound, as the attack cost/loss depends on both the Markov game and the agent's learning algorithms. However, we can consider a special case of attacking V-learning to discuss the tightness of the bounds in Theorem 9. V-learning uses an adversarial bandit method to choose actions, which has a regret lower bound scaling on  $\Omega(\sqrt{K})$. Consider a Markov game where the policy that maximize the attackers' rewards is the unique {NE, CE, CEE} of the original Markov game. The attacker does not need to attack. In this case, there exists an white-box attacker whose attack cost is 0 and attack loss scales on $\Omega(\sqrt{K})$. The bounds in Theorem 9 scales on $\Omega({K}^{2/3})$. The proposed black-box attack's cost and loss are sub-linear but are not optimal. It is because that we use a exploration-then-attack method. An adaptive attack method could potentially achieve optimal cost/loss dependency on $\sqrt{K}$. Finding the optimal attack strategy is an interesting future direction for us to pursue.

---

> > ### Comment · Reviewer_qHBd · 2023-08-20
> >
> > I thank the authors for their comments. However,  I still have concerns regarding the problem of detection and tightness of the bounds. Furthermore, I understand that the numerical results are in the appendix, but given the nature of the topic, I believe it is better to move the results in the main body of the paper.

---

### Official Review · Reviewer_YKVp · 2023-07-06

**Soundness:** 3 good
**Presentation:** 3 good
**Contribution:** 2 fair
**Rating:** 6
**Confidence:** 2

**Summary:**

This paper investigates the impact of adversarial attacks on Multi-Agent Reinforcement Learning (MARL) models. The authors propose an attacker who can manipulate rewards and actions to guide the agents into a target policy or maximize cumulative rewards under a specific reward function. The paper presents an adversarial attack model where the attacker aims to force the agents to learn a target policy or maximize cumulative rewards. Loss and cost functions are used to evaluate the effectiveness of the attack, with cost representing the cumulative manipulation and loss measuring the deviation from the target policy or regret compared to the attacker's optimal policy. The attack problem is studied in three settings: white-box, gray-box, and black-box. The paper demonstrates the limitations of action poisoning-only attacks and reward poisoning-only attacks. Certain Markov Games (MGs) are identified where these strategies are inefficient. However, sufficient conditions are provided under which these attacks can efficiently target MARL algorithms. Efficient strategies for action poisoning and reward poisoning attacks are introduced, and their costs and losses are analyzed. Then a mixed attack strategy is proposed in the gray-box setting, and an approximate mixed attack strategy is introduced for the black-box setting. These strategies can force sub-linear-regret MARL agents to choose actions according to the attacker's target policy with sub-linear cost and loss. The impact of the approximate mixed attack strategy on V-learning, a decentralized MARL algorithm, is investigated. No experiments have been done.

**Strengths:**

Originality: Firstly, this paper introduces an adversarial attack model specifically tailored for Multi-Agent Reinforcement Learning (MARL) systems. While adversarial attacks in single-agent RL have been studied before, this paper focuses on the challenges and implications of attacks in the context of MARL. The consideration of both action poisoning and reward poisoning, as well as the introduction of a mixed attack strategy, adds originality to the research.

Quality: The analysis considers different attack settings, providing a comprehensive understanding of the impact of adversarial attacks on MARL algorithms. The paper also provides conditions under which the attack strategies can be effective, contributing to the quality of the research.

Clarity: The paper is easy to follow. The introduction provides an overview of the problem and motivation, while the contributions are explicitly listed, making it easy for readers to understand what the paper aims to achieve. The attack model, attack settings, and attack strategies are explained in a concise and understandable manner.

Significance: With the increasing use of MARL in various applications, understanding the vulnerabilities and countermeasures against adversarial attacks is crucial for the safe and reliable deployment of these systems. By investigating the limitations of existing attack strategies and proposing new attack strategies, the paper sheds light on the security aspects of MARL and contributes to the development of more robust and trustworthy MARL algorithms.

**Weaknesses:**

Limited comparison with existing work: The paper lacks a comprehensive comparison with prior research on adversarial attacks in MARL. While it briefly mentions existing works on attacks in single-agent RL and MARL, it does not provide a detailed comparison or highlight the novelty and differentiation of its proposed attack model and strategies. A more extensive discussion and comparison with related works would strengthen the paper's contribution and contextualize its findings within the existing literature.

No empirical evaluation: The paper would benefit from an empirical evaluation of the proposed attack strategies. While the analysis provides theoretical insights into the effectiveness of the attack strategies, it lacks practical validation through experiments on real-world or simulated MARL environments. Conducting experiments and presenting empirical results would enhance the credibility and applicability of the proposed attack strategies.

**Questions:**

Some experiments would benefit the paper. It would be great if the authors can resolve my concerns in the weakness.

---

> ### Author Rebuttal · Authors · 2023-08-10
>
> > "Limited comparison with existing work"
>
> Thank you very much for your comment. We will add the following discussion to the revised paper.
>
> Adversarial attacks on single agent RL have been studied in various settings [Behzadan and Munir, 2017, Huang and Zhu, 2019, Ma et al., 2019, Zhang et al., 2020b, Sun et al., 2021, Rakhsha et al., 2020, 2021, Rangi et al., 2022]. Among the existing works on attacks in single-agent RL, the most related paper is [Rangi et al., 2022], which studies the limitations of reward only manipulation or action only manipulation in single-agent RL and proposed an attack strategy  combining reward and action manipulation. There are multiple differences between our work and [Rangi et al., 2022]. First, the MARL is modeled as a Markov game, but the single-agent RL is modeled as a MDP. In Markov game, each agent's action will impact other agents' rewards. Second, the learning object of single-agent RL and MARL is different. The single-agent RL algorithms learn the optimal policy, but MARL algorithms learn the equilibrium. Since the attacks on one agent will impact all other agents and the equilibrium is considered as the agents' learning object, we have to develop techniques to carefully analyze the impact of attacks and the bound of the attack cost. For example, we developed the value function difference decomposition in equations (22)-(27), (29)-(38), (41)-(43), and (51), which build a connection between the value function difference and the number of times that the non-target actions are chosen in MARL setting.
>
> [Ma et al., 2022] considers a game redesign problem where the designer knows the full information of the game and can redesign the reward functions. The proposed redesign methods can incentivize players to take a specific target action profile frequently with a small cumulative design cost.  Ma's work considered the norm-form game but we considered the Markov game. The norm-form game is a simple case of the Markov game with $H = 1$. [Gleave et al., 2020, Guo et al., 2021] study the poisoning attack on multi agent reinforcement learners, assuming that the attacker controls one of the learners. In our work, the attacker is not one of the learners, but an external unit out of the original Markov game. The attacker can poisoning the reward/action of all learners at the same time so that can fool the learners to learn a specific policy. [Wu et al., 2022] studies the reward poisoning attack on offline multi-agent reinforcement learners. The attacker can poisoning the reward of the agents. We considered the online MARL. In offline MARL, the attacker can estimate the underline Markov game from the offline datasets. In online MARL, the attacker may not have the knowledge (reward/transition functions) of the Markov game.
>
> > "No empirical evaluation."
>
> Due to the page limitation, we put the Numerical results in Appendix B. In particular, in Section B.1, we empirically compared the performance of the action poisoning only attack strategy ($d$-portion attack), the reward poisoning only attack strategy ($\eta$-gap attack) and the mixed attack strategy. We considered two different target policies. In Case 1, any action poisoning only attack and reward poisoning only attack will fail. In Case 2, condition 1 and 2 hold so that $d$-portion attack and $\eta$-gap attack work.
> Furthermore, in Section B.2, we considered a synthetic environment and empirically compared the performance of the mixed attack strategy and the approximate mixed attack strategy.
>
> In addition, in the author rebuttal pdf file, we also added more experimental results to discuss how the attack loss and cost are affected by the parameters.

---

> > ### Comment · Reviewer_YKVp · 2023-08-15
> > **Thank you for your response**
> >
> > I would thank the authors for their detailed responses to my questions. I have no concerns now and would like to raise my rating from 5 to 6. Though I got to know the experiments are in the appendix, I still suggest putting some experiment results in the main text when preparing the camera-ready version.

---

> > > ### Author Response · Authors · 2023-08-17
> > > **Thanks!**
> > >
> > > We thank the reviewer again for the thoughtful comments and valuable recommendations. We will definitely add some experiment results in the main text of the revised version.

---

### Official Review · Reviewer_SPgM · 2023-07-08

**Soundness:** 3 good
**Presentation:** 4 excellent
**Contribution:** 3 good
**Rating:** 6
**Confidence:** 4

**Summary:**

This paper studies adversarial attacks in online multi-agent RL, focusing on reward and action poisoning attacks. It provides a set of characterization results for three different attack modalities: white-box, gray-box and black-box attacks. The authors first discuss the  limitation of action-only or reward-only poisoning attack, showing that there are instances of the problem setting where these are not efficient and successful. However, for white-box versions of these attacks, the authors provide sufficient conditions that enable feasibility. For all three attack modalities, they demonstrate that the combination of action and reward poisoning attacks can always force targeted behavior, and provide upper bounds on the expected cost and loss of proposed attack strategies.

**Strengths:**

- The paper is clearly written and is enjoyable to read. While there are typos, these should be easy to fix. More importantly, the technical content is clearly conveyed, and the paper provides intuitions bending the main technical results.
- The paper complements prior work on poisoning attacks in MARL, which considered offline RL setting. Hence, the characterization results are novel and contribute to the line of work on poisoning attacks in RL. That said, the results appear to be similar to the ones presented in [Rangi et al., 2022]. It would be useful to have additional discussions on how these differ in terms of their technical content, proof techniques, etc..
- The paper considers several variations of poisoning attacks under different attack modalities, while showcasing the existence of efficient and successful reward+action poisoning attack strategies. Such a systematic study is likely to be valuable to researchers working on poisoning attacks in RL.
- The paper provides a good overview of the related work on adversarial attacks in RL. Specifically for MARL, the most important references seem to be covered. However, there are a couple of recent references that could be included in the list (see below). I encourage the authors to do so.

> - Wang et al;, BACKDOORL: Backdoor Attack against Competitive Reinforcement Learning
> - Mohammadi et al., Implicit Poisoning Attacks in Two-Agent Reinforcement Learning: Adversarial Policies for Training-Time Attacks
> - Liu et al., Controllable Attack and Improved Adversarial Training in Multi-Agent Reinforcement Learning
> - Chen et al., Byzantine-Robust Online and Offline Distributed Reinforcement Learning


**Weaknesses:**

- In terms of results, the paper does not discus the optimality of the upper bounds in their formal results. For example, it could be interesting to discuss the tightness of the bounds in Theorem 9, and how they compare to those of the underlying learning algorithms (V-learning).
-  Following up on my previous remark, additional discussions regarding some of the assumptions in the problems setting would be useful to have. For example, the paper assumes that the target policy receives strictly positive rewards (page 4). Additionally, it would be good to formally specify the learners' model, and reference it in Theorem 1 and 2.
- This work is primarily theoretical, but the paper could benefit from having additional experiments (in addition to the numerical simulations reported in the appendix), e.g., those that would demonstrate the efficacy of the black-box approach. Namely, bound in (8) and (9) are inversely proportional to $R_{min}$, so it would be interesting to investigate how this dependency affects the practicality of the proposed approach.
- Practical considerations are not discussed, e.g., it would be useful to have some discussion on how to enable scalability and go beyond the tabular setting.

**Questions:**

It would be great if the authors could additionally answer the following clarification questions:
- Could you explain how these results compare to the ones from [Rangi et al., 2022] in terms of, e.g., proof techniques?
- For Eq. (1), Theorem 1 and 2, what is the assumed model of the MARL agents? I.e., what is needed for these statements to hold?
- Can you explain why the analysis does not allow sparse rewards for the target policy (the assumption on page 4)? What happens if the rewards are equal to $0$?
- In the section that talks about reward-only white-box attacks, why doesn't it suffice to set the reward to the lowest value whenever the target action is not taken?
- The dependency on $m$ seems to be smaller in gray-box than in white-box attacks. Could you comment on the practical benefits of white-box attacks?
- Could you comment the optimality of the upper bounds (attack loss/cost)? More specifically, could you comment on the tightness of the bounds in Theorem 9, e.g., in terms of horizon $H$?

**Limitations:**

This paper provides a theoretical analysis of poisoning attacks in multi-agent RL, so I don't believe this work will have any negative societal impact. The limitations of this work could have been discussed in greater detail, e.g., in the concluding remarks.  For example, instead of summarize the results of the paper, the concluding remarks could focus more on the tightness of the bounds in Theorem 9, or the practical aspects of this work, e.g., how to go beyond the tabular setting.

---

> ### Author Rebuttal · Authors · 2023-08-10
>
> > "Specifically for MARL, the most important references seem to be covered. However, there are a couple of recent references that could be included in the list (see below). I encourage the authors to do so."
>
> Thank you very much for bringing these very interesting papers to our attention. We will include these recent related works in the revised version.
>
> > "In terms of results, the paper does not discuss the optimality of the upper bounds in their formal results. For example, it could be interesting to discuss the tightness of the bounds in Theorem 9, and how they compare to those of the underlying learning algorithms (V-learning)."
>
> Currently, we are not able to provide a information-theoretic lower bound, as the attack cost/loss depends on both the Markov game and the agent's learning algorithms. However, we can consider a special case of attacking V-learning to discuss the tightness of the bounds in Theorem 9.  V-learning uses an adversarial bandit method to choose actions, which has a regret lower bound scaling on  $\Omega(\sqrt{K})$. Consider a Markov game where the target policy that maximizes the attackers' rewards is the unique {NE, CE, CEE} of the original Markov game. The attacker does not need to attack. In this case, there exists a white-box attacker whose attack cost is $0$ and attack loss scales on $\Omega(\sqrt{K})$. The bounds in Theorem 9 scales on $\Omega({K}^{2/3})$. The proposed black-box attack's cost and loss are sub-linear but are not optimal. It is because that we use a exploration-then-attack method. An adaptive attack method could potentially achieve optimal cost/loss dependency on $\sqrt{K}$. Finding the optimal attack strategy is an interesting future direction for us to pursue.
>
> > "Following up on my previous remark, additional discussions regarding some of the assumptions in the problems setting would be useful to have. For example, the paper assumes that the target policy receives strictly positive rewards (page 4). Additionally, it would be good to formally specify the learners' model, and reference it in Theorem 1 and 2."
>
> The assumption that the target policy receives strictly positive rewards is essential for the success of the gray-box attack. The attack cost/loss of the gray-box attack scale on $O(m \mathcal{R}(T)/R_{min})$. If the target policy receives zero rewards and $R_{min} = 0$, the gray-box attack does not work.
>
> In Theorem 1 and 2, we do not limit the learners' model, as the object in Equation (1) only includes the restriction of the post-attacked Markov game. The agents do not need to be an efficient learner. However, the object in Equation (1) implies that the learners are rational and can converge to one of NE, CE or CEE if any exists. As action poisoning only attacks and the reward poisoning only attacks can not always change the target policy to a NE, CE or CEE, so the rational learner can not converge to the target policy.
>
> We will add these discussions in the revised paper.
>
> > "This work is primarily theoretical, but the paper could benefit from having additional experiments (in addition to the numerical simulations reported in the appendix), e.g., those that would demonstrate the efficacy of the black-box approach."
>
> Thanks for the suggestion. We added more experimental results to discuss how the attack loss and cost are affected by the parameters in the author rebuttal pdf file.
>
> > "Practical considerations are not discussed, e.g., it would be useful to have some discussion on how to enable scalability and go beyond the tabular setting."
>
> The gray-box attack strategies can be directly used in large-scale environments, even in some high-dimensional continuous environment.  However, in the continuous space, the attacker does not change the non-target action to $0$ but to $r_{i.h}*e^{c |a_{i,h}-a^+_{i,h}|}$, in order to avoiding sparse reward. The ideas of the black-box attack strategies still work. However, the exploration phase should resort to some function approximation methods to efficiently explore an approximate target policy. Then the attack phase keeps the same. We will add these discussions to the revised paper.
>
> >  "Could you explain how these results compare to the ones from [Rangi et al., 2022] in terms of, e.g., proof techniques?"
>
> We added more discussion about our work with the existing work. Due to the rebuttal length limit, we put the discussion into the author rebuttal to every reviewers.
>
> > "In the section that talks about reward-only white-box attacks, why doesn't it suffice to set the reward to the lowest value whenever the target action is not taken?"
>
> For a normal-form game, the method suggested by you will work. However, it does not always work in Markov games. An example is provided in Appendix D.2. The intuitive idea is that reducing the current reward of the non-target action may not impact the agent's choice as the long-term reward of the non-target action could be huge.
>
> > "The dependency on $m$ seems to be smaller in gray-box than in white-box attacks. Could you comment on the practical benefits of white-box attacks?"
>
> The proposed white-box attack strategies are the action poisoning only attacks and the reward poisoning only attacks, which only manipulate the reward or action. The proposed gray-box attack strategy (mixed attack strategy) requires the ability of manipulating the action and reward at the same time. The mixed attack strategy also works in white-box case. However, in some situations where the attacker can only manipulate the reward or action, the mixed attack strategy does not work but the proposed white-box attack strategies work.
>
> > More specifically, could you comment on the tightness of the bounds in Theorem 9, e.g., in terms of horizon?
>
> Currently, we are not able to provide a formal information-theoretic lower bound, as the attack cost/loss depends on both the Markov game and the agent's learning algorithms. It is an important future direction for us to pursue.

---

> > ### Comment · Reviewer_SPgM · 2023-08-20
> > **Thank you for your response**
> >
> > Thank you for your detailed response and for answering my questions. It would be great if you could update the paper accordingly. I don't have further clarification questions.

---

### Author Rebuttal · Authors · 2023-08-10

We thank the reviewers for their valuable feedback, suggestions, and time invested. Here, we answer the reviewers' common concerns.

**Related works.** Due to the page limit of the main paper, we do not provide a comprehensive comparison with prior research on adversarial attacks. We will add the following discussion to the revised paper.

Among the existing works on attacks in single-agent RL, the most related paper is [Rangi et al., 2022], which studies the limitations of reward only manipulation or action only manipulation in single-agent RL and proposed an attack strategy  combining reward and action manipulation.

There are multiple differences between our work and [Rangi et al., 2022]. First, the MARL is modeled as a Markov game, but the single-agent RL is modeled as a MDP. In Markov game, each agent's action will impact other agents' rewards. Second, the learning object of single-agent RL and MARL is different. The single-agent RL algorithms learn the optimal policy, but MARL algorithms learn the equilibrium. Since the attacks on one agent will impact all other agents and the equilibrium is considered as the agents' learning object, we have to develop techniques to carefully analyze the impact of attacks and the bound of the attack cost. For example, we developed the value function difference decomposition in equations (22)-(27), (29)-(38), (41)-(43), and (51), which build a connection between the value function difference and the number of times that the non-target actions are chosen in MARL setting.

Here, we discuss the related work on the adversarial attacks on single MARL.  [Ma et al., 2019] studies reward poisoning attack against batch RL in which the attacker is able to gather and modify the collected batch data. [Rakhsha et al., 2020] proposes a white-box environment poisoning model in which the attacker could manipulate the original MDP to a poisoned MDP. [Behzadan and Munir, 2017, Zhang et al., 2020., Rangi et al., 2022] study online white-box reward poisoning attacks in which the attacker could manipulate the reward signal before the agent receives it. [Sun et al., 2021] proposes a practical black-box poisoning algorithm called VA2C-P. Their empirical results show that VA2C-P works for deep policy gradient RL agents without any prior knowledge of the environment. [Rakhsha et al., 2021] develops a black-box reward poisoning attack strategy called U2, that can provably attack any efficient RL algorithms. [Xu et al., 2021] investigates training-time attacks on RL agents and the introduced attacker can manipulate the environment.

Here, we discuss the related work on the adversarial attacks on MARL. [Ma et al., 2022] considers a game redesign problem where the designer knows the full information of the game and can redesign the reward functions. The proposed redesign methods can incentivize players to take a specific target action profile frequently with a small cumulative design cost.  Ma's work considered the norm-form game but we considered the Markov game. The norm-form game is a simple case of the Markov game with $H = 1$. [Gleave et al., 2020, Guo et al., 2021] study the poisoning attack on multi agent reinforcement learners, assuming that the attacker controls one of the learners. In our work, the attacker is not one of the learners, but an external unit out of the original Markov game. The attacker can poisoning the reward/action of all learners at the same time so that can fool the learners to learn a specific policy. [Wu et al., 2022] studies the reward poisoning attack on offline multi-agent reinforcement learners. The attacker can poisoning the reward of the agents. We considered the online MARL. In offline MARL, the attacker can estimate the underline Markov game from the offline datasets. In online MARL, the attacker may not have the knowledge (reward/transition functions) of the Markov game.  [Wang et al., 2021] studies the backdoor attack in two-player competitive RL systems. The trigger is the action of another agent in the environment. They propose a unified method to design fast-failing agents which will fast fail when trigger occurred. [Liu et al., 2022] studies the controllable attack by constraining the state distribution shift caused by the adversarial policy and offering a more controllable attack scheme. [chen et al., 2022] considers a situation that $\alpha$-fraction of agents are adversarial and can report arbitrary fake information. They design two Byzantine-robust distributed  value iteration algorithms that can identify a near-optimal policy with near-optimal sample complexity. [Mohammadi et al., 2023] studies targeted poisoning attacks in a two-agent setting where an attacker implicitly poisons the effective environment of one of the agents by modifying the policy of its peer.

**Experimental results**  Due to the page limitation, we put the Numerical results in Appendix B. In particular, in Section B.1, we empirically compared the performance of the action poisoning only attack strategy ($d$-portion attack), the reward poisoning only attack strategy ($\eta$-gap attack) and the mixed attack strategy. We considered two different target policies. In Case 1, any action poisoning only attack and reward poisoning only attack will fail. In Case 2, condition 1 and 2 hold so that $d$-portion attack and $\eta$-gap attack work. Furthermore, in Section B.2, we considered a synthetic environment and empirically compared the performance of the mixed attack strategy and the approximate mixed attack strategy. In addition, in the author rebuttal pdf file, we also added more experimental results to discuss how the final attack loss and cost (after $10^6$ episodes) are affected by the parameters $R_{min}$. Noted that the attack loss in mix-attack strategy (gray-box) is the count of the non-target arm defined in Loss1, but the attack loss in approximation mix-attack strategy (black-box) is the regret of the attacker's reward defined in Loss2.

---

### Decision · Program_Chairs · 2023-09-21

**Decision:**

Accept (poster)

**Comment:**

In this paper the authors investigate adversarial poisoning attacks against online multi-agent RL algorithms, where the attacker can either manipulate reward or action feedbacks.

In particular, they look at three different attack scenarios: white-box, gray-box and black-box attacks. The authors first show that action-only or reward-only poisoning attacks are not efficient in gray-box and back-box settings. On the other hand, for white-box case, the authors provide sufficient conditions that enable successful manipulations. For all three attack scenarios, they demonstrate that the combination of action and reward poisoning attacks can always succeed, and provide upper bounds on the expected cost and loss of proposed attack strategies.

After the rebuttal phase, perhaps the only remaining concern is that the numerical results were deferred to the appendix, and several reviewers suggested to move some of the main numerical evaluations into the main paper, subject to acceptance and space availability. I find this concern to be minor. However, I also strongly recommend the authors to revise the paper to comply with this suggestion, should the paper be accepted.